# DiffE2E: Rethinking End-to-End Driving with a Hybrid Diffusion-Regression-Classification Policy

**Rui Zhao**[1][*] **Yuze Fan**[1] **Ziguo Chen**[1] **Fei Gao**[1,2][†] **Zhenhai Gao**[1,2]

[1]College of Automotive Engineering, Jilin University

[2]National Key Laboratory of Automotive Chassis Integration and Bionics, Jilin University

https://infinidrive.github.io/DiffE2E/

## Abstract

End-to-end learning has emerged as a transformative paradigm for autonomous driving. However, the inherently multimodal nature of driving behaviors remains a fundamental challenge to robust deployment. We propose **DiffE2E**, a diffusion-based end-to-end autonomous driving framework. The architecture first performs multi-scale alignment of perception features from multiple sensors via a hierarchical bidirectional cross-attention mechanism. Subsequently, we design a hybrid diffusion-regression-classification decoder based on the Transformer architecture, adopting a collaborative training paradigm to seamlessly fuse the strengths of diffusion and explicit strategies. DiffE2E conducts structured modeling in the latent space: diffusion captures the multimodal distribution of future trajectories, while regression and classification act as explicit strategies to precisely model key control variables such as velocity, enhancing both the precision and controllability of the model. A global condition integration module further enables deep fusion of perception features with high-level goals, significantly improving the quality of trajectory generation. The subsequent cross-attention mechanism facilitates efficient interaction between integrated features and hybrid latent variables, promoting joint optimization of diffusion and explicit strategies for structured output generation and thereby yielding more robust control. Experimental results demonstrate that DiffE2E achieves state-of-the-art performance on both CARLA closed-loop benchmarks and NAVSIM evaluations. The proposed unified framework that integrates diffusion and explicit strategies provides a generalizable paradigm for hybrid action representation and shows substantial potential for extension to broader domains, including embodied intelligence.

## 1 Introduction

End-to-end autonomous driving frameworks directly map sensor data to control commands, effectively avoiding error accumulation inherent in traditional modular pipelines and significantly enhancing both decision-making efficiency and scenario adaptability [21, 27, 6, 47, 5, 48]. Most existing approaches adopt explicit policy architectures based on direct regression [42, 9, 53, 23, 62, 43], learning the mapping between environmental observations and vehicle actions from large-scale driving datasets. However, due to the inherently multimodal nature of driving behaviors, these methods are prone to producing suboptimal solutions [34] and often fail to generate the diverse trajectories consistent with real-world driving strategies.

The multimodal nature of driving decisions means multiple plausible actions may exist for the same scenario. Conventional regression-based approaches tend to average over this diversity, often

---

[*]Contact me at rzhao@jlu.edu.cn

[†]Correspondence to: Fei Gao (gaofei123284123@jlu.edu.cn)

39th Conference on Neural Information Processing Systems (NeurIPS 2025).

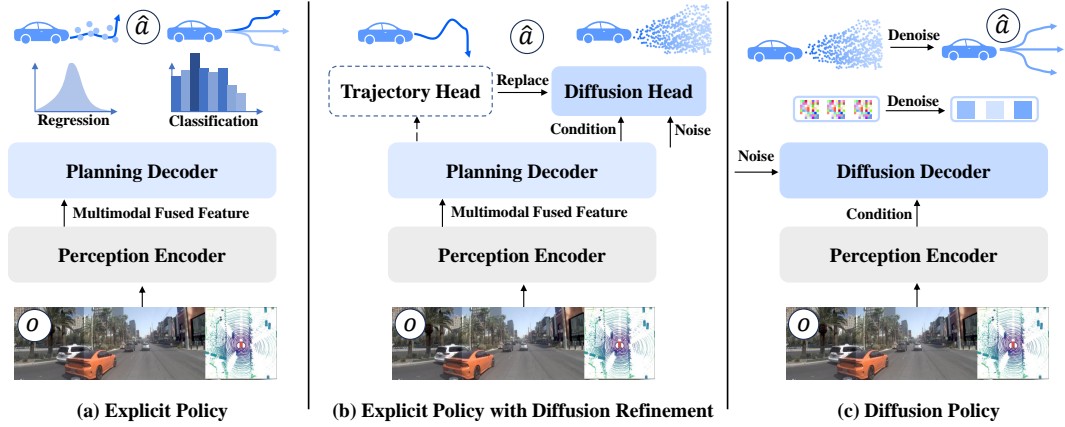

Figure 1: **Comparison of end-to-end training paradigms.** (a) **Explicit Policy.** Directly predicts trajectories through regression after sensor input processing. (b) **Explicit Policy with Diffusion Refinement.** Uses diffusion models to replace traditional explicit policy trajectory output heads. (c) **Diffusion Policy.** Uses diffusion models to directly generate trajectories based on perception encoder features.

resulting in suboptimal or even unsafe behaviors [12]. This issue exacerbates safety risks and limits the reliability of regression-based policies in real-world autonomous driving systems. To address these limitations, recent research [6, 33, 31, 41] has explored explicitly modeling multimodality via predefined discrete sets of trajectory candidates. However, such hard-coded schemes undermine the continuity of policy generation, reduce model adaptability, and ultimately degrade continuous decision making into a fixed mode selection problem, thereby weakening generalization capability.

Diffusion models have demonstrated strong capabilities in modeling complex multimodal distributions and generating high-quality outputs, making them increasingly prominent in computer vision [18, 44, 45, 13]. This generation paradigm based on gradual denoising provides a new solution for addressing multimodal modeling and generalization problems in end-to-end autonomous driving. The successful application of diffusion models in robotic motion planning has already demonstrated their advantages in multimodal action sequence generation and long-term sequence prediction [7, 57, 24]. However, applying diffusion models to autonomous driving systems faces unique challenges: autonomous driving needs to simultaneously meet multiple stringent requirements in open road environments, dealing with the uncertainty of highly dynamic traffic participants while ensuring real-time response; generating feasible trajectories that conform to road topology while ensuring traffic efficiency.

Recent works have explored diffusion models in autonomous driving planning [61] and end-to-end control [34, 54], applying Denoising Diffusion Implicit Model (DDIM) [44], DPM-Solver [36], and Rectified Flow [35] for trajectory generation, revealing the strong potential of generative approaches. However, most integrate diffusion models only after planning decoders, using them to replace explicit policy heads (see Figure 1(b)). This setup risks losing key perception features and constrains generation due to pre-processed decoder outputs. While some methods [34] use trajectory anchors to enhance real-time performance, anchor-based designs can limit trajectory diversity. A more effective architecture is needed to fully harness the generative power of diffusion models.

To address the aforementioned challenges, we introduce **DiffE2E**, an innovative end-to-end autonomous driving framework, as illustrated in Figure 2. DiffE2E first performs multi-scale alignment of LiDAR and image features via a hierarchical bidirectional cross-attention mechanism, thereby enabling high-precision environmental perception [9, 23]. Building upon this, we design a hybrid diffusion-regression-classification decoder based on the Transformer architecture and employ a collaborative training paradigm to seamlessly integrate the strengths of diffusion and explicit strategies. By combining diffusion models with explicit strategies, we structurally partition the latent space: on the one hand, diffusion models are leveraged to capture the distribution, diversity, and higher-order uncertainty of future trajectories; on the other hand, explicit strategies are utilized to achieve fine-grained modeling of key control variables such as velocity. The cross-attention mechanism facilitates efficient interaction between integrated features and hybrid latent variables, supporting the joint optimization of diffusion and explicit strategies for structured output generation. DiffE2E

achieves state-of-the-art performance on both the CARLA closed-loop benchmark and the NAVSIM evaluation, demonstrating enhanced safety and traffic efficiency in complex scenarios, as well as strong generalization capability.

In summary, our contributions are as follows:

1. We propose DiffE2E, an end-to-end autonomous driving framework that uses diffusion models for trajectory generation and is the first to evaluate them in closed-loop tests within CARLA.

2. We propose a hybrid diffusion-regression-classification decoder based on the Transformer architecture, employing a collaborative training paradigm to achieve seamless integration of the advantages of diffusion and explicit strategies.

3. We conduct dual-platform benchmark testing, achieving state-of-the-art performance across multiple benchmarks in the CARLA simulator, and attaining a PDMS of 92.7 in the non-reactive simulation NAVSIM, while maintaining higher real-time performance compared to other methods.

## 2 Preliminaries

**Problem Definition:** This research focuses on end-to-end autonomous driving closed-loop control strategies based on diffusion models. The system directly takes multi-modal raw perception data as input, including front-view camera RGB images $\mathbf{I}_t \in \mathbb{R}^{H \times W \times 3}$, LiDAR point clouds $\mathbf{P}_t^{3D} \in \mathbb{R}^{N \times 3}$, and vehicle state information $\mathbf{s}_t \in \mathbb{R}^{d_s}$. The system predicts the ego vehicle's future trajectory $\boldsymbol{x}_0$, where the generative process is modeled as:

$$p_\theta(\boldsymbol{x}_0|\mathcal{C}) = \int_{\boldsymbol{x}_{1:T}} p(\boldsymbol{x}_T) \prod_{t=1}^{T} p_\theta(\boldsymbol{x}_{t-1}|\boldsymbol{x}_t, \mathcal{C}) \, \mathrm{d}\boldsymbol{x}_{1:T} \tag{1}$$

In diffusion modeling, $\boldsymbol{x}_t$ represents the intermediate variable at step $t$ in the diffusion process, with the final predicted trajectory being $\boldsymbol{x}_0 \in \mathbb{R}^{K \times d_c}$. Here, each waypoint $\mathbf{p}_i \in \mathbb{R}^{d_c}$ represents position information in the predicted trajectory. The condition information $\mathcal{C}$ is encoded from multi-modal sensor data through a cross-modal feature fusion module. Unlike traditional open-loop control, in closed-loop control, the trajectory decision at the current moment directly affects the perception input at the next moment, forming a dynamic feedback loop. This coupled relationship requires the model to have strong temporal consistency and robustness.

**Diffusion Models:** Denoising Diffusion Probabilistic Models (DDPM) [18] are powerful generative models that capture complex multimodal distributions via a two-phase process: forward diffusion gradually adds noise, while the reverse process reconstructs the data through iterative denoising. This framework naturally models the multimodality of driving behaviors. The forward process follows a Markov chain that transforms data $\boldsymbol{x}_0$ into noise over $T$ steps:

$$q(\boldsymbol{x}_t|\boldsymbol{x}_{t-1}) = \mathcal{N}(\boldsymbol{x}_t; \sqrt{1-\beta_t}\boldsymbol{x}_{t-1}, \beta_t\boldsymbol{I}) \tag{2}$$

where $\{\beta_t\}_{t=1}^{T}$ controls noise levels. Through reparameterization, we can sample $\boldsymbol{x}_t$ directly:

$$\boldsymbol{x}_t = \sqrt{\bar{\alpha}_t}\boldsymbol{x}_0 + \sqrt{1-\bar{\alpha}_t}\boldsymbol{\epsilon}, \quad \boldsymbol{\epsilon} \sim \mathcal{N}(0, \boldsymbol{I}) \tag{3}$$

where $\bar{\alpha}_t = \prod_{i=1}^{t}(1-\beta_i)$. While DDPM generates high-quality samples, its sequential process is computationally expensive. DDIM [44] addresses this with a non-Markovian process that accelerates generation while maintaining quality, making diffusion models more practical for real-time autonomous driving applications.

## 3 Methodology

**Overview:** DiffE2E is an end-to-end autonomous driving framework, as illustrated in Figure 2. In the perception stage, it introduces a multimodal spatiotemporal fusion module, which employs a hierarchical bidirectional cross-attention mechanism to align LiDAR and camera features and establish a structured representation of the driving scene. During the decoding stage, a Transformer-based

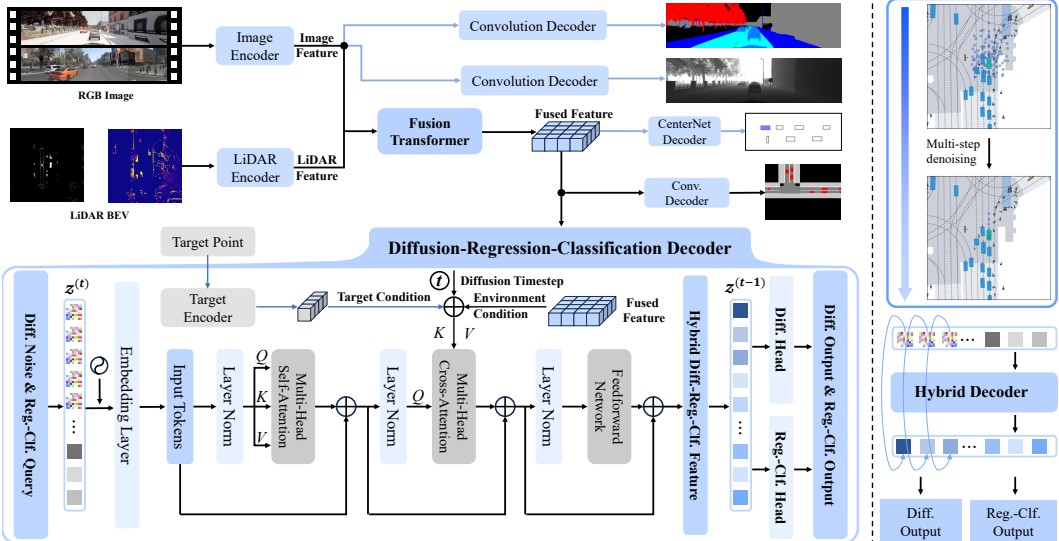

Figure 2: **Overall architecture of DiffE2E.** The main architecture consists of a Transformer-based perception module and a hybrid diffusion-regression-classification decoder. The blue arrows ($\rightarrow$) indicate the data flow exclusively used for the CARLA benchmark, while the **black arrows** ($\rightarrow$) represent the data flow shared between both the CARLA and NAVSIM benchmarks.

hybrid diffusion-regression-classification decoder is adopted, in conjunction with a collaborative training strategy that seamlessly integrates diffusion and explicit policy paradigms. The global condition integration module fuses scene representations with goal-oriented global context, while the cross-attention mechanism enables efficient interaction between integrated features and hybrid latent variables, thereby facilitating the joint generation of diffusion-based trajectories and explicit outputs.

### 3.1 Multimodal Fusion Perception Module

The perception module aims to fuse multimodal sensor data to construct structured environmental representations. This paper adopts the TransFuser architecture as the basic perception backbone network [23], with inputs including wide-angle front-view RGB images $\mathbf{I}_t \in \mathbb{R}^{H \times W \times 3}$ and BEV representations $\mathbf{P}_t^{2D} \in \mathbb{R}^{H \times W \times C}$ constructed from raw LiDAR point clouds $\mathbf{P}_t^{3D} \in \mathbb{R}^{N \times 3}$. After extracting initial features from both branches, they enter a multi-scale cross-fusion module composed of multiple Transformer layers, achieving deep alignment and information interaction between LiDAR and image features through cross-modal attention mechanisms. Finally, the module outputs high-dimensional fusion features $\mathcal{C}$, global semantic representations, and image feature grids to support the fine-grained modeling requirements of downstream decision modules.

### 3.2 Hybrid Diffusion-Regression-Classification Module

After the multimodal fusion perception module integrates information from diverse sensors, the DiffE2E framework introduces an innovative architecture. By incorporating a Transformer-based hybrid diffusion-regression-classification decoder and adopting a collaborative training paradigm, DiffE2E effectively combines the strengths of both diffusion-based and explicit policy strategies, as illustrated in Figure 2. This section provides a comprehensive exposition of the global condition integration, hybrid diffusion-regression-classification decoding process, and decoder output module.

**Global Condition Integration:** To enhance the influence of target points in trajectory generation, they are used as global conditions [23]. Original target points $g$ are first projected via a linear layer $f_{\text{goal}}$ into a shared high-dimensional space of dimension $d$, forming representations $\mathbf{g} \in \mathbb{R}^{\ell_g \times d}$. Meanwhile, diffusion timesteps $t$ are encoded into temporal embeddings $\mathbf{t}_{\text{emb}} \in \mathbb{R}^{\ell_t \times d}$ to help the model adapt across denoising stages. Finally, perception features $\mathcal{C} \in \mathbb{R}^{\ell_c \times d}$, goal features, and timestep embeddings are fused, and combined with learnable positional encoding $\mathbf{E}_{\text{pos}}^{\mathcal{C}}$ to form a contextual representation for trajectory decoding, where $\ell_c$, $\ell_g$, and $\ell_t$ denote the sequence lengths of

conditional features, goal representations, and timestep embeddings:

$$\tilde{\mathcal{C}} = \left[ \text{Concat}(\mathcal{C}, \mathbf{g}, \mathbf{t}_{\text{emb}}) + \mathbf{E}_{\text{pos}}^{\mathcal{C}} \right] \in \mathbb{R}^{(\ell_c + \ell_g + \ell_t) \times d} \tag{4}$$

This global condition integration mechanism incorporates target point information and timestep embeddings into perception features, enhancing the model's awareness of navigation goals and enabling dynamic adjustment of feature representations during denoising for more precise trajectory generation.

**Hybrid Diffusion-Regression-Classification Decoding:** Let $\ell_k$ and $\ell_s$ denote the sequence lengths of the noisy trajectory and explicit policy queries, respectively. The proposed hybrid decoder first encodes the noisy trajectory $\boldsymbol{\tau}_t$ into high-dimensional feature representations via a linear projection layer $f_{\text{enc}}$. These are then concatenated with the initial query vectors for explicit policy decoding, $\mathbf{q}_0 \in \mathbb{R}^{\ell_s \times d}$, and augmented with learnable positional embeddings $\mathbf{E}_{\text{pos}}^{\mathcal{Z}} \in \mathbb{R}^{(\ell_k + \ell_s) \times d}$. This yields the decoder's initial input vector:

$$\mathcal{Z}_{\text{in}} = \left[ \text{Concat}(f_{\text{enc}}(\boldsymbol{\tau}_t), \mathbf{q}_0) + \mathbf{E}_{\text{pos}}^{\mathcal{Z}} \right] \in \mathbb{R}^{(\ell_k + \ell_s) \times d}, \quad t \in \{1, \ldots, T\} \tag{5}$$

At each time step $t$ of the entire diffusion process, the input first passes through a multi-head self-attention layer to process the internal feature relationships of $\mathcal{Z}_{\text{in}}$:

$$\mathcal{Z}_{\text{mid}} = \text{SelfAttn}(\mathcal{Z}_{\text{in}}), \quad \mathcal{Z}_{\text{mid}} \in \mathbb{R}^{(\ell_k + \ell_s) \times d} \tag{6}$$

Then, through a cross-attention mechanism, $\mathcal{Z}_{\text{mid}}$ interacts with the conditional features $\tilde{\mathcal{C}}$, producing the final output:

$$\mathcal{Z}_{\text{out}} = \text{CrossAttn}(\mathcal{Z}_{\text{mid}}, \tilde{\mathcal{C}}, \tilde{\mathcal{C}}), \quad \mathcal{Z}_{\text{out}} \in \mathbb{R}^{(\ell_k + \ell_s) \times d}, \tilde{\mathcal{C}} \in \mathbb{R}^{(\ell_c + \ell_g + \ell_t) \times d} \tag{7}$$

where $\mathcal{Z}_{\text{out}}$ denotes the feature vectors output by the decoder, and $\tilde{\mathcal{C}}$ represents the integrated global conditional features. Within the output representation $\mathcal{Z}_{\text{out}}$, the first $\ell_k$ positions correspond to the latent features for diffusion-based trajectory generation, while the remaining $\ell_s$ positions correspond to the latent features for explicit policy outputs.

**Decoder Output Module:** The decoder output module further processes the hybrid features $\mathcal{Z}_{\text{out}} \in \mathbb{R}^{(\ell_k + \ell_s) \times d}$ to realize joint decoding of both diffusion-based and explicit policy strategies. This module employs a feature separation and task-specific decoding approach, decomposing the output features structurally in semantic space:

$$
\begin{aligned}
\mathcal{Z}_{\text{diff}} &= \mathcal{Z}_{\text{out}}[: \ell_k] \in \mathbb{R}^{\ell_k \times d} \\
\mathcal{Z}_{\text{sup}} &= \mathcal{Z}_{\text{out}}[\ell_k : \ell_k + \ell_s] \in \mathbb{R}^{\ell_s \times d}
\end{aligned}
\tag{8}
$$

where $\mathcal{Z}_{\text{diff}}$ represents the high-dimensional latent representations for the diffusion trajectory, and $\mathcal{Z}_{\text{sup}}$ contains structured feature information for explicit policy outputs.

### 3.3 Diffusion-Regression-Classification Collaborative Training Strategy

Building upon the aforementioned hybrid diffusion-regression-classification decoder architecture, we propose a collaborative training paradigm that synergistically integrates diffusion-based generative modeling with explicit policy learning. The core objective of this strategy is to combine the generative capabilities of diffusion models with the precision of explicit policies, achieving complementary advantages from both approaches.

**Diffusion Loss Function:** DiffE2E employs a trajectory reconstruction-based loss function for diffusion generation, which directly optimizes the model's ability to recover the original trajectory from noisy inputs. Using $\mathcal{M}_\theta$ to represent the entire model, the loss function is formulated as:

$$\mathcal{L}_{\text{diff}} = \mathbb{E}_{t \sim \mathcal{U}(1,T), \mathbf{x}_0, \mathbf{x}_t \sim q_t(\mathbf{x}_t | \mathbf{x}_0)} \left[ \left\| \mathbf{x}_0 - \mathcal{M}_\theta(\mathbf{x}_t, t, \tilde{\mathcal{C}}) \right\|_2^2 \right] \tag{9}$$

**Explicit Policy Loss Function:** The explicit policy loss adopts a multi-task composite optimization strategy, enabling precise control of gradient flows and prioritization through task-specific weighting

coefficients. Specifically, $\mathcal{L}_{\text{sup}} = \sum_{i \in \Omega} \lambda_i \cdot \mathcal{L}_i(\mathbf{y}_i, \hat{\mathbf{y}}_i; \theta_i)$, where $\Omega$ denotes the set of all explicit policy tasks, $\lambda_i$ is the weight for task $i$, and $\mathcal{L}_i(\mathbf{y}_i, \hat{\mathbf{y}}_i; \theta_i)$ represents the objective for task $i$, with $\mathbf{y}_i$ and $\hat{\mathbf{y}}_i$ denoting ground-truth labels and predictions, and $\theta_i$ the corresponding network parameters.

For instance, in the velocity prediction task, we construct a multi-class classification model based on semantic stratification, covering four velocity states with clear physical meanings: *brake*, *walking speed*, *low speed*, and *high speed*. The accuracy of classification prediction is optimized via a weighted cross-entropy loss function:

$$\mathcal{L}_{\text{speed}}(\mathbf{y}, \hat{\mathbf{p}}; \mathbf{w}) = -\frac{1}{B} \sum_{n=1}^{B} \sum_{i=1}^{4} w_i \cdot y_{n,i} \log(\hat{p}_{n,i} + \text{eps}), \quad \text{s.t.} \quad \sum_{i=1}^{4} \hat{p}_{n,i} = 1, \hat{p}_{n,i} \geq 0 \quad (10)$$

where $B$ denotes the batch size, $y_{n,i} \in \{0, 1\}$ indicates the ground-truth label (with one-hot encoding) of the $n$-th sample under the $i$-th velocity class, $\hat{p}_{n,i} \in [0, 1]$ is the predicted normalized classification probability, $w_i > 0$ is the class-balancing weight coefficient, and eps is a constant for numerical stability. Further details on explicit policy loss formulations are provided in Appendix A.

# 4    Experiments

**Experiment Setup:** This research is primarily evaluated using the CARLA simulator closed-loop benchmark [14] and the NAVSIM non-reactive simulation benchmark [12]. CARLA offers diverse urban scenes and sensor emulation, with its closed-loop mechanism providing real-time feedback to assess decision quality over long horizons. NAVSIM, built on OpenScene [11] (a streamlined version of the nuPlan [1] dataset), provides $360°$ coverage via 8 cameras and 5 LiDARs, with 2Hz annotations of maps and object bounding boxes. Additional details are provided in Appendix B.

## 4.1    Experiment on CARLA

**Implementation Details:** This study uses RegNetY-3.2GF [39] as the encoder for image and LiDAR inputs. To optimize computation and training efficiency, a two-stage strategy is adopted: the first trains the perception module with multi-task losses; the second trains the diffusion decoder conditioned on the frozen perception outputs. We adopt CARLA Longest6, CARLA Town05 Long, and CARLA Town05 Short as evaluation benchmarks [9, 38], using the official Driving Score (DS), Route Completion (RC), and Infraction Score (IS) as metrics. Detailed implementation details and baseline descriptions are provided in Appendix B.1.

**Main Results:** As shown in Table 1, our proposed diffusion-based method, DiffE2E, demonstrates superior performance on the CARLA Longest6 benchmark, consistently outperforming mainstream existing approaches. Across the three key evaluation metrics, DiffE2E achieves a DS of $79 \pm 4$, RC of $94 \pm 2$, and IS of $0.84 \pm 0.02$, each exceeding or closely matching the best results reported to date. Notably, compared to the strong baseline TF++ (which also utilizes RegNetY-3.2GF as the backbone for perception and the same input modalities), DiffE2E delivers a 10-point improvement in DS ($79 \pm 4$ vs. $69 \pm 0$), a 0.12 increase in IS ($0.84 \pm 0.02$ vs. $0.72 \pm 0.01$), and comparable stability in RC ($94 \pm 2$ vs. $94 \pm 2$), thoroughly validating the dual advantages of our model in both accuracy and safety. Furthermore, the standard deviation ($\pm$) of DiffE2E is similar to other methods, indicating robust and stable performance across multiple experimental runs. Traditional explicit-policy-based methods (e.g., LAV v2, TransFuser) generally exhibit lower DS and IS when facing complex scenarios, revealing their limitations in perceiving and reasoning about multimodal driving behaviors. In contrast, diffusion decoding significantly enhances the model's generalization and decision diversity capabilities. In summary, DiffE2E achieves higher DS and IS, ensuring both high route completion rates and low infraction rates over long driving horizons, and demonstrates exceptional end-to-end closed-loop autonomous driving performance.

**Visualization:** Figure 3 shows a comparison in a typical right-turn scenario. Initially, both TF++ and DiffE2E plan similar paths by first merging right. When a vehicle appears, TF++ sticks to its preset path and collides, while DiffE2E adapts by temporarily going forward, then safely merging after the vehicle passes. This demonstrates DiffE2E's superior multimodal generation capability and real-time adaptability in dynamic traffic, effectively avoiding collisions.

Table 1: **Comparison on the CARLA Longest6 benchmark.** "C & L" denotes the use of both camera and LiDAR as sensor inputs. "Pri." indicates the use of privileged information. The **best** and second best results are highlighted in **bold** and underline.

| Method | Traj. Decoder | Img. Encoder | Input | DS ↑ | RC ↑ | IS ↑ |
|---|---|---|---|---|---|---|
| Expert | - | - | - | 81 ± 3 | 90 ± 1 | 0.91 ± 0.04 |
| WOR [4] | - | ResNet-34 [17] | C | 21 ± 3 | 48 ± 4 | 0.56 ± 0.03 |
| LAV v1 [2] | Explicit Policy | ResNet-18 [17] | C&L | 33 ± 1 | 70 ± 3 | 0.51 ± 0.02 |
| InterFuser [42] | Explicit Policy | ResNet-50 [17] | C&L | 47 ± 6 | 74 ± 1 | 0.63 ± 0.07 |
| TransFuser [9] | Explicit Policy | RegNetY-3.2GF [39] | C&L | 47 ± 6 | 93 ± 1 | 0.50 ± 0.06 |
| TCP [53] | Explicit Policy | ResNet-34 [17] | C | 48 ± 3 | 72 ± 3 | 0.65 ± 0.04 |
| ThinkTwice [26] | Explicit Policy | ResNet-50 [17] | C&L | 51 ± 4 | 64 ± 4 | 0.80 ± 0.03 |
| LAV v2 [2] | Explicit Policy | ResNet-18 [17] | C&L | 58 ± 1 | 83 ± 1 | 0.68 ± 0.02 |
| Perception PlanT [40] | Explicit Policy | RegNetY-3.2GF [39] | C&L | 58 ± 5 | 88 ± 1 | 0.65 ± 0.06 |
| DriveAdapter [25] | Explicit Policy | ResNet-50 [17] | C&L | 58 ± 2 | 73 ± 3 | 0.79 ± 0.04 |
| TF++ [23] | Explicit Policy | RegNetY-3.2GF [39] | C&L | 69 ± 0 | **94 ± 2** | 0.72 ± 0.01 |
| **DiffE2E (Ours)** | Diffusion Policy | RegNetY-3.2GF [39] | C&L | **79 ± 4** | **94 ± 2** | **0.84 ± 0.02** |

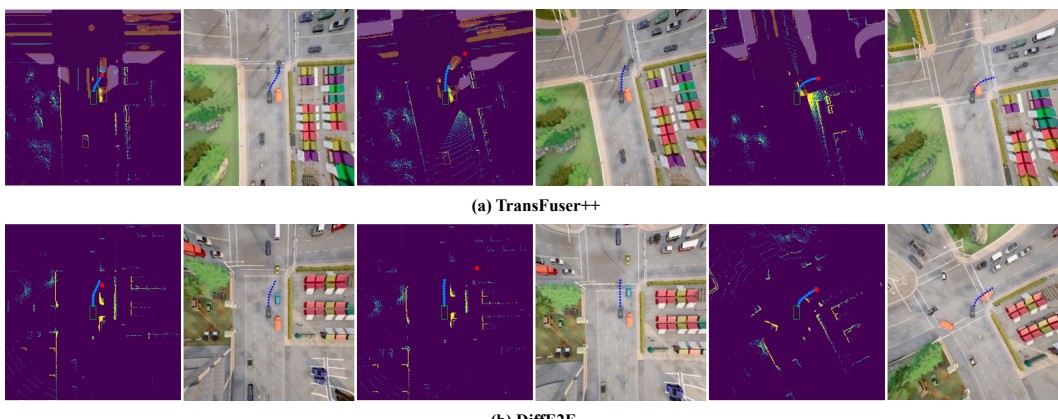

(a) TransFuser++

(b) DiffE2E

Figure 3: **Visualization in CARLA Simulator.** In both the LiDAR and scene visualizations, blue points represent the predicted trajectory, while red points in the LiDAR view denote the target waypoints.

## 4.2 Experiment on NAVSIM

**Implementation Details:** This study builds a model training framework based on NAVSIM's navtrain dataset. Unlike the CARLA setup, we adopt VovNetV2-99 [30] as the feature extraction backbone network in NAVSIM. The Predictive Driver Model Score (PDMS) is used as a comprehensive metric, combining key driving dimensions via weighted integration: No at-fault Collision (NC), Drivable Area Compliance (DAC), Time-To-Collision (TTC), Comfort (C), and Ego Progress (EP). Detailed implementation details and baseline descriptions can be found in Appendix B.2.

**Main Results:** As shown in Table 2, DiffE2E achieves outstanding performance on the NAVSIM benchmark, attaining a PDMS score of 92.7—surpassing Hydra-MDP++ (91.0), Hydra-MDP$^\diamond$ (91.0), GoalFlow (90.3), and DiffusionDrive (88.1). This result highlights the superiority of our hybrid diffusion-based end-to-end approach across multiple driving metrics. In terms of safety and compliance, DiffE2E particularly excels: it achieves a faultless collision rate of 99.9 (compared with Hydra-MDP++ at 98.6, Hydra-MDP$^\diamond$ at 98.7, and GoalFlow at 98.4), and reaches the highest drivable area compliance score of 98.6, on par with Hydra-MDP++. For the time-to-collision metric, DiffE2E leads with a score of 99.3, outperforming Hydra-MDP++ by 4.2 points. In terms of efficiency and comfort, DiffE2E achieves an ego progress of 85.3, just below Hydra-MDP++ (85.7) and comparable to Hydra-MDP$^\diamond$ (86.5). Additionally, its comfort score reaches 99.9, nearly matching the best-performing methods, demonstrating its ability to generate smooth, human-like trajectories.

Unlike DiffusionDrive and GoalFlow, which utilize diffusion as post-processing after the planning decoder, DiffE2E employs a full diffusion decoding paradigm, enabling richer policy space modeling and greater expressive power. Furthermore, the hybrid design of diffusion and explicit policy

Table 2: **Comparison on the Navtest benchmark.** "S" denotes using only the ego vehicle state as input. * indicates that these two methods adopt diffusion models as post-planning decoders, whereas DiffE2E employs a diffusion model as the full decoder to directly generate trajectories. † and ‡ denote the variants of DiffE2E with the image encoder replaced by ResNet-34 and with the sensor input restricted to images only, respectively. ◇ denotes that the result is obtained via sub-score ensembling of three backbones [33].

| Method | Traj. Decoder | Img. Enc. | Input | PDMS↑ | NC↑ | DAC↑ | EP↑ | TTC↑ | C↑ |
|--------|---------------|-----------|-------|-------|-----|------|-----|------|-----|
| Human | - | - | - | 94.8 | 100 | 100 | 87.5 | 100 | 99.9 |
| AD-MLP [58] | Explicit Policy | - | S | 65.6 | 93.0 | 77.3 | 62.8 | 83.6 | **100** |
| VADv2 [6] | Explicit Policy | ResNet-34 [17] | C&L | 80.9 | 97.2 | 89.1 | 76.0 | 91.6 | **100** |
| UniAD [21] | Explicit Policy | ResNet-34 [17] | C | 83.4 | 97.8 | 91.9 | 78.8 | 92.9 | **100** |
| LTF [9] | Explicit Policy | ResNet-34 [17] | C | 83.8 | 97.4 | 92.8 | 79 | 92.4 | **100** |
| PARA-Drive [52] | Explicit Policy | ResNet-34 [17] | C | 84.0 | 97.9 | 92.4 | 79.3 | 93 | 99.8 |
| TransFuser [9] | Explicit Policy | ResNet-34 [17] | C&L | 84.0 | 97.7 | 92.8 | 79.2 | 92.8 | **100** |
| LAW [32] | Explicit Policy | ResNet-34 [17] | C | 84.6 | 96.4 | 95.4 | 81.7 | 88.7 | 99.9 |
| DRAMA [56] | Explicit Policy | ResNet-34 [17] | C&L | 85.5 | 98 | 93.1 | 80.1 | 94.8 | **100** |
| Hydra-MDP [33] | Explicit Policy | ResNet-34 [30] | C&L | 86.5 | 98.3 | 96.0 | 78.7 | 94.6 | **100** |
| DiffusionDrive [34] | Diffusion Policy* | ResNet-34 [17] | C&L | 88.1 | 98.2 | 93.7 | 82.2 | 94.7 | **100** |
| GoalFlow [54] | Diffusion Policy* | V2-99 [30] | C&L | 90.3 | 98.4 | 98.3 | 85 | 94.6 | **100** |
| Hydra-MDP◇ [33] | Explicit Policy | ViT-L [15] + V2-99 [30] | C&L | 91.0 | 98.7 | 98.2 | **86.5** | 95.0 | **100** |
| Hydra-MDP++ [31] | Explicit Policy | V2-99 [30] | C | 91.0 | 98.6 | **98.6** | 85.7 | 95.1 | **100** |
| **DiffE2E†** | Diffusion Policy | ResNet-34 [17] | C&L | 89.8 | 99.2 | 96.8 | 83.6 | 96.7 | **100** |
| **DiffE2E‡** | Diffusion Policy | V2-99 [30] | C | 90.9 | 99.7 | 97.1 | 84.2 | 98.2 | 99.9 |
| **DiffE2E (Ours)** | Diffusion Policy | V2-99 [30] | C&L | 92.7 | **99.9** | **98.6** | 85.3 | **99.3** | 99.9 |

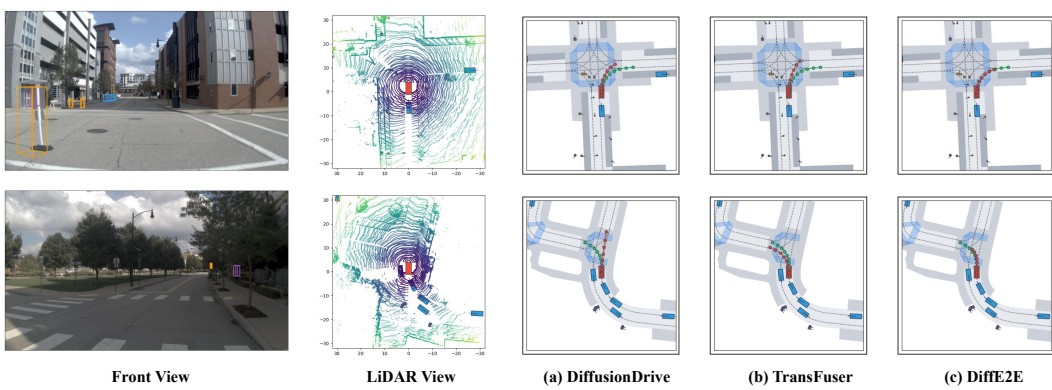

| | | | | |
|--|--|--|--|--|
| **Front View** | **LiDAR View** | **(a) DiffusionDrive** | **(b) TransFuser** | **(c) DiffE2E** |

Figure 4: **Visualization in Navtest benchmark.** Red trajectories denote the predicted paths of each method, while green trajectory corresponds to the ground truth.

(regression and classification) combines the strengths of both approaches: it captures the multimodal distribution of future trajectories while maintaining the accuracy and controllability of explicit policies. To further validate the impact of key design choices, we evaluate two variants: DiffE2E† (with a ResNet-34 encoder, PDMS 89.8) and DiffE2E‡ (using image input only, PDMS 90.9). The performance degradation of these variants underscores the importance of high-capacity visual encoders and multi-modal sensor fusion. Overall, DiffE2E achieves state-of-the-art or near-optimal results on all critical NAVSIM metrics, demonstrating robust, controllable, and efficient performance, and offering a new paradigm for end-to-end autonomous driving.

**Visualization:** To validate the generalization and superiority of DiffE2E, we selected two representative complex driving scenarios for comparative analysis (Figure 4). Green trajectories denote human references, and red ones indicate planned trajectories. In right-turn intersections, baseline methods often deviate or cross boundaries, while DiffE2E accurately follows lane edges with smooth turns. In small intersection left-turns, DiffusionDrive misinterprets navigation intent and plans a straight trajectory, TransFuser incorrectly chooses the right lane, while only DiffE2E accurately executes the left-turn instruction with a trajectory almost completely matching the reference. This demonstrates DiffE2E's accuracy and safety in trajectory planning. More visualizations can be found in Appendix C.

Table 3: Ablation results of DiffE2E on CARLA Longest6 benchmark.

| Type | Method | DS | RC | IS |
|------|--------|----|----|----|
| Base | **DiffE2E** | **78.9** | **93.7** | **0.84** |
| Input | w/o ego state | 68.9 | 88.8 | 0.81 |
|       | w/o command | 69.6 | 93.8 | 0.77 |
| Component | w/o GRU | 66.8 | 75.5 | 0.88 |
| Training Paradigm | Full Diffusion | 70.1 | 91.1 | 0.76 |
|                   | Full Discrimination | 70.3 | 83.5 | 0.84 |
|                   | One-Stage Training | 18.2 | 21.8 | 0.79 |
|                   | **Two-Stage Training** | **78.9** | **93.7** | **0.84** |
| Output | Noise Prediction | 20.1 | 27.2 | 0.72 |
|        | **Trajectory Prediction** | **78.9** | **93.7** | **0.84** |

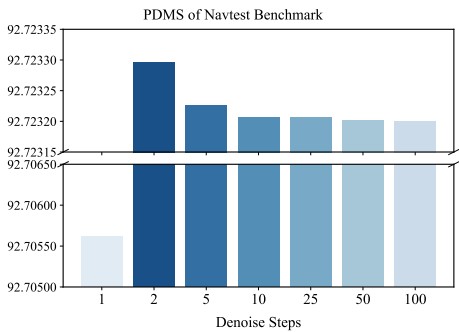

Figure 5: Ablation study of denoising steps on Navtest benchmark.

## 4.3 Ablation Studies

To systematically evaluate the contribution of each component in the DiffE2E framework, we conducted ablation experiments on the CARLA Longest6 benchmark (see Table 3). The results show that, at the input level, removing the ego vehicle state or navigation command leads to a significant drop in driving score from 78.9 to 68.9 and 69.6, respectively, confirming the critical roles of ego state for precise planning and navigation command for intent understanding. At the architectural level, removing the GRU module reduces the score to 66.8, indicating its indispensability in modeling temporal dependencies in complex scenarios. Regarding training paradigm, using only the diffusion strategy (70.1) or the discriminative strategy (70.3) underperforms the hybrid design, demonstrating the effectiveness of combining both. Notably, single-stage training achieves only 18.2 points, 77% lower than two-stage training, and performs very poorly. This is because direct coupling of perception and planning in single-stage training makes it difficult for the model to simultaneously ensure perception accuracy and planning rationality, causing significant conflicts between optimization objectives. This manifests as the vehicle hardly being able to stay in the lane, with frequent yawing, deviation or even collisions. For output types, when the diffusion model predicts noise, the score drops dramatically to 20.1 and the vehicle fails to follow the lane, while directly predicting trajectories provides the best results. This suggests that, for high-precision driving tasks, modeling structured trajectory space is clearly more effective. For further discussion on output mechanisms, please refer to Appendix D.1.

Additionally, we conducted an ablation study on the number of denoising steps in the diffusion model (see Figure 5). Because the CARLA environment is highly stochastic and the impact of denoising step count is relatively small, we adopt the more stable Navtest benchmark for analysis. Experimental results show that PDMS achieves the lowest score at one step (insufficient denoising), peaks at two steps, and then gradually declines, indicating that two steps are sufficient for denoising. This benefits from our temporal-spatial decoupled hybrid modeling strategy and directly aligning the diffusion objective to trajectory error, allowing the model to focus more on learning the spatial distribution structure. Consequently, high-quality outputs can be produced with fewer iterations.

## 4.4 Real-time Performance Comparison of DiffE2E

To comprehensively evaluate the real-time performance of DiffE2E, we conducted inference latency comparisons on the NAVSIM benchmark using a single NVIDIA RTX 3090 GPU, against existing diffusion-based methods (DiffusionDrive and GoalFlow) and the non-diffusion method TransFuser. Results are presented in Table 4. The experiments show that TransFuser, equipped with the lightweight ResNet-34 image encoder, achieves the lowest latency (29.1ms), followed by DiffusionDrive (39.1ms). In contrast, DiffE2E and GoalFlow, both using the larger

Table 4: Comparison of inference latency and model parameters between different methods.

| Method | Latency (ms) ↓ | Params | Img. Enc. |
|--------|---------------|--------|-----------|
| TransFuser [12] | 29.1 | 55M | ResNet-34 [17] |
| DiffusionDrive [34] | 39.1 | 60M | ResNet-34 [17] |
| GoalFlow [54] | 66.8 | 190M | V2-99 [30] |
| **DiffE2E (Ours)** | 42.8 | 105M | V2-99 [30] |

VoVNetV2-99 encoder, report latencies of 42.8ms and 66.8ms, respectively. Notably, inference latency exhibits a clear positive correlation with model parameter scale, reflecting the inherent trade-off between model complexity and computational efficiency. Nevertheless, with an optimized diffusion sampling strategy, DiffE2E achieves real-time performance comparable to lightweight models while maintaining a relatively high parameter count (105M), fully validating its balanced optimization between computational efficiency and model expressiveness.

## 5 Related Works

**End-to-end Autonomous Driving:** End-to-end autonomous driving has advanced significantly in multimodal perception fusion and decision-making. UniAD [21] constructs a full-stack Transformer to coordinate perception-prediction-planning tasks, VAD [27] designs vectorized scene representation to improve planning efficiency, VADv2 [6] models action-space distribution through a trajectory vocabulary library, SparseDrive [47] proposes sparse trajectory representation for efficient driving without BEV, and the Hydra-MDP series [33, 31] designs a multi-teacher distillation framework to integrate rule-based systems with human driving knowledge. TransFuser [9] fuses camera and LiDAR features via Transformer for intersection decisions; TCP [53] jointly trains trajectory and control predictions; InterFuser [42] introduces safety thinking maps for multi-view multimodal fusion, and TF++ [23] enhances the decoder and proposes decoupled speed prediction. However, such policy-based approaches often compress complex multimodal driving behaviors into a single deterministic output, which tends to yield averaged, less diverse suboptimal solutions in varied decision-making scenarios, making it difficult to fully capture the richness and uncertainty inherent in driving behaviors.

**Diffusion Model in Transportation and Autonomous Driving:** Diffusion models are profoundly transforming the transportation and autonomous driving fields with their excellent multimodal generation capabilities. In transportation, Diffusion-ES [55] innovatively combines evolutionary strategies with diffusion models, achieving complex driving behavior generation without requiring differentiable reward functions, with its zero-shot performance significantly surpassing traditional methods in the nuPlan benchmark. VBD [22] uses game theory to guide adversarial scenario generation, enhancing simulation realism. MotionDiffuser [28] introduces a permutation-invariant architecture for constrained multi-agent trajectory sampling, ensuring interaction consistency. Diffusion Planner [61] leverages DPM-Solver [36] and classifier guidance for fast, safe, and personalized trajectory generation in closed-loop planning. However, most of these methods are based on perfect perception assumptions, ignoring the impact of state estimation errors caused by perception uncertainties in practical applications. In the field of end-to-end autonomous driving, although the application of diffusion models has achieved preliminary results, it still faces many challenges. DiffusionDrive [34] first introduced diffusion for end-to-end driving with an anchored strategy. HE-Drive [50] uses conditional DDPM [18] and vision-language models for scoring, producing human-like, spatiotemporally consistent trajectories at high computational cost. GoalFlow [54] addresses trajectory divergence via goal-driven flow matching and efficient one-step generation. These works demonstrate the enormous potential of diffusion models in the field of end-to-end autonomous driving.

## 6 Conclusion

This study presents an innovative end-to-end autonomous driving framework, DiffE2E, which features a hybrid diffusion-regression-classification decoder based on Transformer architecture and introduces a collaborative training mechanism to effectively integrate the strengths of diffusion models and explicit policies. We design a structured latent space modeling approach: the diffusion model captures the diversity and uncertainty of driving behaviors by modeling the future trajectory distribution, while the explicit policy precisely models key control variables such as ego-vehicle speed to enhance controllability and prediction accuracy. This hybrid paradigm not only increases the model's fault tolerance through multi-step denoising generation but also leverages the structured generation mechanism and implicit ensemble reasoning to improve robustness against distribution shifts. In both CARLA closed-loop testing and NAVSIM non-interactive simulation, DiffE2E achieves state-of-the-art performance, balancing traffic efficiency and safety, and demonstrates strong generalization potential.

## Acknowledgments and Disclosure of Funding

This work was supported by the National Natural Science Foundation of China (grant 52572475) and the Science and Technology Development Project of Jilin Province (grant 20250102130JC).

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

# A  Details of Multi-task Loss Function Design

## A.1  Loss Function Design

Compared to the deterministic mapping learning paradigm of explicit policies, diffusion models require denoising at multiple noise levels, which introduces greater optimization complexity and computational overhead. To improve training efficiency and ensure stable model convergence in the CARLA simulation environment, we construct a hierarchical two-stage training strategy: the first stage focuses on optimizing the parameters of the multi-modal perception module, while the second stage adopts an end-to-end joint training paradigm to optimize both the perception module and the diffusion decoder parameters.

In the first stage, we adopt a multi-task learning framework and design task-specific objectives for core perception sub-tasks, including high-precision semantic segmentation, monocular depth estimation, and 3D object detection [23]. An adaptive weighting mechanism is used to construct a composite loss function, enabling efficient parameter learning for the perception module. Furthermore, this work proposes a modular loss function system, with each component defined as follows:

- **Image Semantic Segmentation Loss** $\mathcal{L}_{\text{sem}}$: Supervises semantic segmentation predictions from the image perspective, using a class-weighted cross-entropy loss function:

$$\mathcal{L}_{\text{sem}}(\mathbf{y}, \hat{\mathbf{y}}; \mathbf{w}) = -\sum_{c=1}^{C} w_c \cdot \sum_{i=1}^{H \times W} y_{i,c} \log(\hat{y}_{i,c} + \text{eps}) \tag{11}$$

  where $C$ is the total number of semantic classes, $H \times W$ is the image resolution, $y_{i,c} \in \{0, 1\}$ indicates whether pixel $i$ belongs to class $c$ in the ground truth, $\hat{y}_{i,c} \in [0, 1]$ is the predicted probability, $w_c > 0$ is the balancing weight for class $c$, and eps is a constant for numerical stability.

- **Bird's-Eye View Semantic Segmentation Loss** $\mathcal{L}_{\text{bev}}$: For semantic predictions in BEV space, cross-entropy loss is calculated only within the camera-visible region $\mathcal{M}_{\text{valid}}$:

$$\mathcal{L}_{\text{bev}}(\mathbf{S}_{\text{bev}}, \hat{\mathbf{S}}_{\text{bev}}; \mathcal{M}_{\text{valid}}) = -\frac{1}{|\mathcal{M}_{\text{valid}}|} \sum_{(x,y) \in \mathcal{M}_{\text{valid}}} \sum_{c=1}^{C} S_{\text{bev}}(x,y,c) \log(\hat{S}_{\text{bev}}(x,y,c) + \text{eps}) \tag{12}$$

  where $\mathcal{M}_{\text{valid}}$ represents the set of valid pixels, $|\mathcal{M}_{\text{valid}}|$ is the number of valid pixels, $S_{\text{bev}}(x,y,c)$ and $\hat{S}_{\text{bev}}(x,y,c)$ are the ground truth label and predicted probability for class $c$ at BEV coordinate $(x, y)$, respectively.

- **Depth Estimation Loss** $\mathcal{L}_{\text{depth}}$: As an auxiliary task for the image branch, masked $\ell_1$ loss is used for depth supervision:

$$\mathcal{L}_{\text{depth}}(\mathbf{D}, \hat{\mathbf{D}}; \mathcal{M}_{\text{depth}}) = \frac{1}{|\mathcal{M}_{\text{depth}}|} \sum_{(u,v) \in \mathcal{M}_{\text{depth}}} |D(u,v) - \hat{D}(u,v)| \tag{13}$$

  where $\mathcal{M}_{\text{depth}}$ is the set of valid depth pixels, $D(u,v)$ and $\hat{D}(u,v)$ are the ground truth depth value and predicted depth value at pixel $(u, v)$, respectively.

- **Object Detection Loss** $\mathcal{L}_{\text{det}}$: This loss function combines multiple subtasks to jointly optimize key attributes of objects, including position, size, orientation, velocity, and braking status, to enhance overall detection accuracy and temporal consistency. It includes weighted combinations of several subtask loss functions. Each subloss function is defined as follows:

  * **Center Heatmap Loss** $\mathcal{L}_{\text{hm}}$: Uses Gaussian focal loss to optimize object center point prediction:

$$\mathcal{L}_{\text{hm}}(\mathbf{P}, \hat{\mathbf{P}}; \alpha, \gamma) = \frac{1}{N} \sum_{(x,y) \in \Omega} \begin{cases} (1 - \hat{P}_{x,y})^\alpha \log(\hat{P}_{x,y}), & \text{if } P_{x,y} = 1 \\ (1 - P_{x,y})^\gamma (\hat{P}_{x,y})^\alpha \log(1 - \hat{P}_{x,y}), & \text{otherwise} \end{cases} \tag{14}$$

    where $N$ is the number of valid objects, $\Omega$ is the feature map space, $P_{x,y}$ and $\hat{P}_{x,y}$ are the ground truth heatmap value and predicted heatmap value at position $(x, y)$, respectively, and $\alpha$ and $\gamma$ are focal loss modulation parameters.

* **Object Size Loss** $\mathcal{L}_{\text{wh}}$: Uses $\ell_1$ loss to optimize object width and height prediction:

$$\mathcal{L}_{\text{wh}}(\mathbf{W}, \hat{\mathbf{W}}) = \frac{1}{N} \sum_{i=1}^{N} \|\mathbf{W}_i - \hat{\mathbf{W}}_i\|_1 \tag{15}$$

where $\mathbf{W}_i = [w_i, h_i]^T$ and $\hat{\mathbf{W}}_i = [\hat{w}_i, \hat{h}_i]^T$ are the ground truth size and predicted size of the $i$-th object, respectively.

* **Center Point Offset Loss** $\mathcal{L}_{\text{off}}$: Also uses $\ell_1$ loss to optimize fine-grained center point position:

$$\mathcal{L}_{\text{off}}(\mathbf{O}, \hat{\mathbf{O}}) = \frac{1}{N} \sum_{i=1}^{N} \|\mathbf{O}_i - \hat{\mathbf{O}}_i\|_1 \tag{16}$$

where $\mathbf{O}_i = [o_{x,i}, o_{y,i}]^T$ and $\hat{\mathbf{O}}_i = [\hat{o}_{x,i}, \hat{o}_{y,i}]^T$ are the ground truth offset and predicted offset of the $i$-th object, respectively.

* **Yaw Classification Loss** $\mathcal{L}_{\text{cls}}$: Uses cross-entropy loss to optimize yaw angle category prediction:

$$\mathcal{L}_{\text{cls}}(\mathbf{Y}, \hat{\mathbf{Y}}) = \frac{1}{N} \sum_{i=1}^{N} -\log\left(\frac{\exp(\hat{Y}_{i,c_i})}{\sum_{j=1}^{K} \exp(\hat{Y}_{i,j})}\right) \tag{17}$$

where $c_i$ is the ground truth yaw angle category of the $i$-th object, $K$ is the total number of yaw angle categories, and $\hat{Y}_{i,j}$ is the predicted score of the $i$-th object belonging to the $j$-th category.

* **Yaw Residual Loss** $\mathcal{L}_{\text{res}}$: Uses smooth $\ell_1$ loss to optimize yaw angle residual prediction:

$$\mathcal{L}_{\text{res}}(\mathbf{R}, \hat{\mathbf{R}}; \delta) = \frac{1}{N} \sum_{i=1}^{N} \text{smooth}_{L1}(R_i - \hat{R}_i; \delta) \tag{18}$$

where $R_i$ and $\hat{R}_i$ are the ground truth yaw angle residual and predicted yaw angle residual of the $i$-th object, respectively, and $\text{smooth}_{L1}$ is the smooth $\ell_1$ loss function. The smooth L1 loss function is defined as:

$$\text{smooth}_{L1}(z; \delta) = \begin{cases} 0.5 \cdot z^2/\delta, & \text{if } |z| < \delta \\ |z| - 0.5 \cdot \delta, & \text{otherwise} \end{cases} \tag{19}$$

The above loss functions form the core components of the first-stage training for the perception module. By adopting a multi-task learning paradigm, the model can simultaneously reconstruct multiple scene feature representations, including semantic segmentation maps, depth maps, and bird's-eye view segmentation. This approach not only achieves efficient and comprehensive scene understanding but also lays a solid perceptual foundation for subsequent trajectory generation tasks, ensuring that downstream decision-making modules can perform precise reasoning based on high-quality scene representations.

Additionally, on the NAVSIM benchmark, we use agent detection as an auxiliary task to enhance the model's perception capabilities in complex driving scenarios. The agent detection task adopts a hierarchical loss structure, including two complementary subtasks: existence discrimination and geometric parameter regression:

* **Agent Existence Classification Loss:** Through binary cross-entropy loss, the model's ability to determine the presence of agents in the scene is optimized:

$$\mathcal{L}_{\text{exist}}(\mathbf{y}, \hat{\mathbf{p}}; \alpha, \beta) = -\frac{1}{N} \sum_{n=1}^{N} \left[ \alpha \cdot y_n \cdot (1 - \hat{p}_n)^\beta \log(\hat{p}_n) + (1 - \alpha) \cdot (1 - y_n) \cdot \hat{p}_n^\beta \log(1 - \hat{p}_n) \right] \tag{20}$$

where $y_n \in \{0, 1\}$ represents the ground truth label of whether an agent exists in the $n$-th sample, $\hat{p}_n \in [0, 1]$ is the predicted existence probability, and $\alpha \in [0, 1]$ and $\beta \geq 0$ are focal loss modulation parameters used to adjust the gradient contribution ratio of easy and difficult samples, increasing the model's attention to difficult samples.

Table 5: Hyperparameters of DiffE2E on CARLA.

| Benchmark | Type | Parameter | Value |
|---|---|---|---|
| CARLA | Non-diffusion | Epochs (One-stage training) | 30 |
| | | Epochs (Two-stage training) | 30 |
| | | Loss weight: diffusion | 1.0 |
| | | Loss weights: semantic, BEV, depth, speed | 1.0 |
| | | Loss weights: center heatmap, size, offset, yaw | 1.0 |
| | | Prediction horizon (Timesteps) | 10 |
| | | Learning rate | 3e-4 |
| | | Batch size (One-stage training) | 256 |
| | | Batch size (Two-stage training) | 16 |
| | | Weight decay | 0.01 |
| | Diffusion | Noise schedule | Square Cosine Schedule |
| | | Noise coefficient | 0.0001, 0.02 |
| | | Number of forward diffusion steps | 100 |
| | | Number of reverse denoising steps | 2 |

- **Agent Geometry Parameter Regression Loss:** For existing agents, an adaptively weighted smooth L1 loss function is used to accurately regress their spatial position and morphological parameters:

$$\mathcal{L}_{\text{box}}(\mathbf{b}, \hat{\mathbf{b}}; \boldsymbol{\mu}, \delta) = \frac{1}{M} \sum_{m=1}^{M} \sum_{i \in \{x,y,w,h,\theta\}} \mu_i \cdot \text{smooth}_{L1}(b_{m,i} - \hat{b}_{m,i}; \delta) \tag{21}$$

where $M$ is the number of samples with agents in the batch, $b_{m,i}$ and $\hat{b}_{m,i}$ represent the ground truth value and predicted value of the $i$-th parameter of the agent bounding box in the $m$-th sample, respectively, $\boldsymbol{\mu} = [\mu_x, \mu_y, \mu_w, \mu_h, \mu_\theta]^T$ is the importance weight vector for each parameter, and $\delta$ is the threshold parameter for the smooth L1 loss. This hierarchical loss design can simultaneously optimize the classification accuracy and localization precision of agent detection, ensuring accurate regression of bounding box parameters while maintaining high recall rates.

# B   Details of Experimental Setup

## B.1   CARLA Benchmarks

### B.1.1   Implementation Details

This study is based on a dataset constructed using the MPC expert from the TF++ framework [23], comprising a total of 750,000 frames of high-quality driving data. We employ the RegNetY-3.2GF network [39] as the encoder for both image and LiDAR inputs. To optimize computational resources and improve training efficiency, we design a two-stage training strategy: the first stage focuses on the perception module, utilizing a multi-task loss function that includes semantic segmentation, depth estimation, object detection bounding boxes, and bird's-eye view prediction. In the second stage, the pre-trained perception module from stage one is loaded and further trained, using its outputs as conditional information to train the diffusion decoder. The entire training is divided into two stages, each trained for 30 epochs, with an initial learning rate of 3e-4. Batch size is adapted for different stages—16 for the first stage and 256 for the second stage—to accelerate the convergence of the diffusion model. The controller used is consistent with that in TF++. The specific hyperparameter settings are shown in Table 5. All experiments are conducted on four NVIDIA 3090 GPUs.

In the CARLA autonomous driving evaluation system, we use three core metrics to measure model performance. First is Route Completion (RC), which quantifies the percentage of the predetermined route completed by the vehicle, calculated as RC $= \frac{D_{\text{traveled}}}{D_{\text{total}}} \times 100\%$, where $D_{\text{traveled}}$ is the actual distance traveled and $D_{\text{total}}$ is the total route length. Second is the Infraction Score (IS), used to evaluate driving safety by accumulating penalty coefficients for violations: IS $= \prod_{i=1}^{N} p_i$, where $p_i$ is the penalty coefficient for the $i$-th violation (such as pedestrian collision 0.50, vehicle collision 0.60, static object collision 0.65, running red lights 0.70, running stop signs 0.80, etc.), and $N$ is the

total number of violations. Finally, the Driving Score (DS) serves as a comprehensive evaluation metric, combining completion and safety: $DS = RC \times IS$. A higher DS value indicates stronger ability to complete tasks while ensuring safety, making it a key indicator for evaluating the overall performance of autonomous driving systems.

### B.1.2 Benchmarks

**Longest6 Benchmark** is a high-difficulty, scalable evaluation benchmark designed to verify the performance of autonomous driving models in complex scenarios. Following [9], we use Longest6 as an alternative evaluation scheme for local ablation studies and multiple experiments. This benchmark selects the 6 longest routes from each town among the 76 training routes provided by CARLA, totaling 36 routes with an average length of 1.5 kilometers (close to the official leaderboard's 1.7 kilometers). To increase evaluation difficulty, we set the highest traffic density, including numerous dynamic obstacles (vehicles, pedestrians), combined with 6 weather conditions (such as Cloudy, Wet, HardRain) and 6 lighting conditions (such as Night, Dawn, Noon), while also including predefined adversarial scenarios (such as obstacle avoidance, unprotected left turns, pedestrians suddenly crossing, etc.). Compared to benchmarks like NoCrash and NEAT routes with shorter routes and lower traffic density, Longest6 more closely resembles the challenges of real scenarios, providing a more comprehensive validation of model robustness in complex urban environments (such as multi-lane intersections and dense traffic).

**Town05 Benchmark** is another important evaluation benchmark in CARLA, divided into Town05 Short and Town05 Long settings. Town05 Short includes 10 short routes (100-500 meters), each with 3 intersections; Town05 Long includes 10 long routes (1000-2000 meters), each with 10 intersections. These two settings are specifically designed to verify model driving capabilities in high-density dynamic traffic environments (including vehicles and pedestrians) and complex scenarios (such as unprotected intersection turns and randomly appearing pedestrians). Typically, models perform better on the Short benchmark, while performance decreases on the Long benchmark, reflecting the challenges of handling complex scenarios and long-term planning in long-distance driving. We use the same evaluation metrics as Longest6 (DS, RC, IS) to measure model performance on the Town05 benchmark.

### B.1.3 Baselines

- *WOR* [4]: A model learning method based on the "world-on-rails" assumption, computing action value functions through offline dynamic programming to achieve driving policy distillation without actual interaction.

- *LAV* [2]: A full-vehicle learning framework that enhances training data diversity through viewpoint-independent BEV representation and trajectory prediction modules utilizing surrounding vehicle trajectories. The authors released two versions: LAV v1 and LAV v2.

- *InterFuser* [42]: A safety-enhanced multi-modal sensor fusion framework that integrates multi-view camera and LiDAR information through Transformers, outputting interpretable safety mental maps to constrain control actions.

- *TransFuser* [9]: A Transformer-based multi-modal fusion architecture that fuses image and LiDAR features through cross-view attention mechanisms, enabling navigation in complex long-distance scenarios.

- *TCP* [53]: A trajectory-control joint prediction framework combining GRU temporal modules and trajectory-guided attention mechanisms, integrating dual-branch outputs through scene-adaptive fusion strategies.

- *PlanT* [40]: An interpretable planning Transformer based on privileged states, achieving global traffic element reasoning through object-centered representation and self-attention mechanisms, with attention weights visualizing decision bases.

- *DriveAdapter* [25]: A perception-planning decoupling framework that aligns student perception features with teacher planning features through learnable adapters, enhancing policy safety through action-guided mask learning.

- *ThinkTwice* [26]: A cascaded decoding paradigm that achieves action iterative optimization through a three-stage "observe-predict-refine" mechanism, utilizing scene feature retrieval and future state prediction.

Table 6: Comparison on the CARLA Town05 Long benchmark.

| Method | Traj. Decoder | Img. Encoder | Input | DS ↑ | RC ↑ | IS ↑ |
|---|---|---|---|---|---|---|
| CILRS [10] | Explicit Policy | ResNet-34 [17] | C | 7.8 | 10.3 | 0.75 |
| LBC [3] | Explicit Policy | ResNet-18,34 [17] | C | 12.3 | 31.9 | 0.66 |
| Roach [60] | Explicit Policy | ResNet-34 [17] | C | 41.6 | 96.4 | 0.43 |
| ST-P3 [20] | Explicit Policy | EfficientNet-B4 [49] | C | 11.5 | 83.2 | - |
| VAD [27] | Explicit Policy | ResNet-50 [17] | C | 30.3 | 75.2 | - |
| MILE [19] | Explicit Policy | ResNet-18 [17] | C | 61.1 | 97.4 | 0.63 |
| DriveMLM [51] | Explicit Policy | ViT-G [59] | C&L | 76.1 | 98.1 | 0.78 |
| VADv2 [6] | Explicit Policy | ResNet-50 [17] | C | 85.1 | 98.4 | 0.87 |
| **DiffE2E (Ours)** | Diffusion Policy | RegNetY-3.2GF [39] | C&L | **90.8** | **100** | **0.91** |

Table 7: Comparison on the CARLA Town05 Short benchmark.

| Method | Traj. Decoder | Img. Encoder | Input | DS ↑ | RC ↑ |
|---|---|---|---|---|---|
| CILRS [10] | Explicit Policy | ResNet-34 [17] | C | 7.5 | 13.4 |
| LBC [3] | Explicit Policy | ResNet-18,34 [17] | C | 31.0 | 55.0 |
| TransFuser [9] | Explicit Policy | RegNetY-3.2GF [39] | C&L | 54.5 | 78.4 |
| ST-P3 [20] | Explicit Policy | EfficientNet-B4 [49] | C | 55.1 | 86.7 |
| NEAT [8] | Explicit Policy | ResNet-34 [17] | C | 58.7 | 77.3 |
| Roach [60] | Explicit Policy | ResNet-34 [17] | C | 65.3 | 88.2 |
| WOR [4] | - | ResNet-18,34 [17] | C | 64.8 | 87.5 |
| VAD [27] | Explicit Policy | ResNet-50 [17] | C | 64.3 | 87.3 |
| VADv2 [6] | Explicit Policy | ResNet-50 [17] | C | 89.7 | 93.0 |
| InterFuser [42] | Explicit Policy | ResNet-50 [17] | C&L | 95.0 | 95.2 |
| **DiffE2E (Ours)** | Diffusion Policy | RegNetY-3.2GF [39] | C&L | **95.2** | **99.7** |

- *TF++* [23]: An improved version of TransFuser that eliminates target point following bias through a Transformer decoder. *TF++* uses explicit uncertainty modeling with path and speed classification, while the *TF++WP* variant retains waypoint prediction, achieving deceleration decisions through longitudinal averaging of continuous waypoints.

### B.1.4 Other Results

In addition to CARLA Longest6, this research also rigorously evaluated the DiffE2E method on the CARLA Town05 Long and Town05 Short benchmarks, with detailed results shown in Table 6 and Table 7. On the Town05 Long benchmark, DiffE2E achieved a DS of 90.8 and an RC of 100, outperforming existing methods. On the Town05 Short benchmark, where environmental complexity is relatively lower, all algorithms generally showed improved performance, with the DiffE2E method still demonstrating excellent results, achieving a DS of 95.2 and an RC of 99.7. The outstanding performance across Town05 scenarios of varying complexity further validates the robustness and generalization capability of our method.

### B.1.5 Analysis of failure cases

This section provides a systematic visual analysis of the ablation study results from Section 4.3, with the aim of professionally and intuitively comparing how different design choices affect the performance of the DiffE2E method. We focus on analyzing two representative types of failure cases: One-Stage Training and Noise Prediction.

As shown in Figure 6, we systematically compare the performance of the diffusion model when using noise versus original trajectory as the output target in the same intersection-left-turn scenario. According to the visualization results, when using noise as the output target, the predicted trajectory distribution tends to be more concentrated during the straight driving stage, but this does not improve future trajectory prediction capability. After entering the intersection, the trajectory predicted based on noise deviates significantly, causing the vehicle to fail to make a correct left turn along the intended path. In contrast, directly using the original trajectory as the prediction target allows the model to

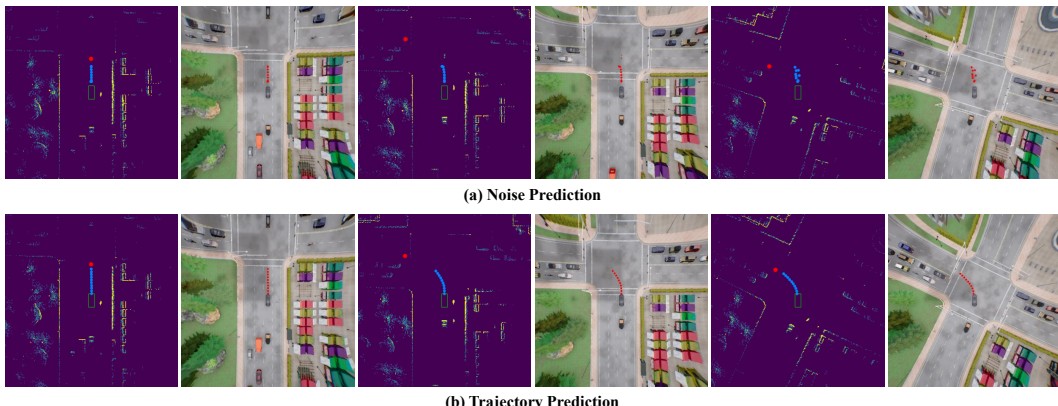

(a) Noise Prediction

(b) Trajectory Prediction

Figure 6: **Visualization of output type ablation experiment.**

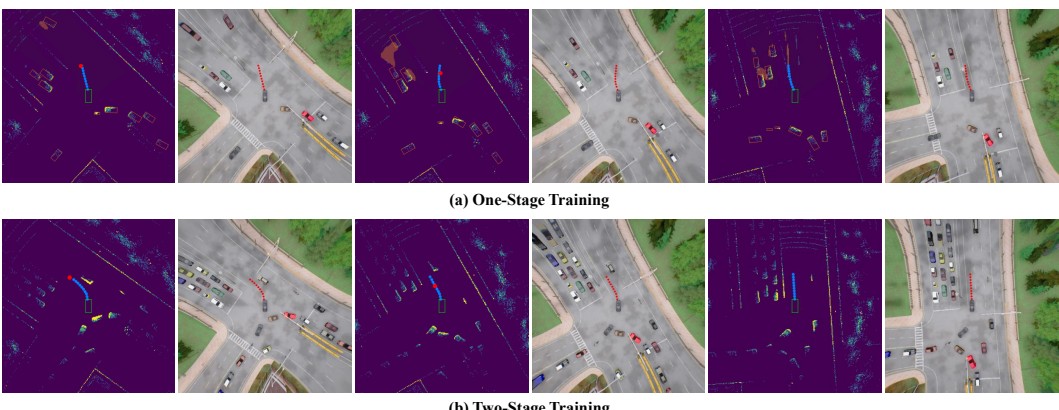

(a) One-Stage Training

(b) Two-Stage Training

Figure 7: **Visualization of training stage ablation experiment.**

better adhere to the left-turn route and complete the left turn successfully, demonstrating superior trajectory generation ability.

Figure 7 systematically compares the performance of one-stage training and two-stage training strategies in the intersection-left-turn scenario. The visualization shows that one-stage training leads to obvious deviations in the ego-vehicle trajectory, making it difficult for the vehicle to smoothly execute the left-turn maneuver. In contrast, with the two-stage training strategy, the generated ego-vehicle trajectory is smoother and better aligned with the desired path, significantly improving decision accuracy and trajectory feasibility in complex intersection scenarios.

## B.2 NAVSIM

### B.2.1 Implementation Details

This research builds a model training framework based on NAVSIM's navtrain dataset. Unlike the image encoding architecture used in the CARLA simulation environment, we adopted VovNetV2-99 [30] as the feature extraction backbone network in our NAVSIM experiments, a choice inspired by the Hydra-MDP [33] method, which can more effectively capture visual features in complex driving scenarios. All experiments were conducted on four NVIDIA 3090 GPUs. The specific hyperparameter settings are shown in Table 8.

The navtrain dataset establishes a highly complex driving environment evaluation system, characterized by carefully selected challenging scenarios with frequently changing driving intentions, deliberately excluding low-complexity driving situations such as stationary states and constant-speed cruising. This evaluation system quantitatively analyzes planning algorithm performance through a combination of non-reactive simulation and closed-loop evaluation mechanisms. This study uses the

Table 8: Hyperparameters of DiffE2E on NAVSIM.

| Benchmark | Type | Parameter | Value |
|-----------|------|-----------|-------|
| NAVSIM | Non-diffusion | Epochs | 100 |
| | | Batch size | 64 |
| | | Prediction horizon (Timesteps) | 8 |
| | | Learning rate | 1e-4 |
| | | Ego progress weight | 5.0 |
| | | Time to collision weight | 5.0 |
| | | Comfort weight | 2.0 |
| | | Loss weights: diffusion | 10.0 |
| | | Loss weights: agent class, agent box, BEV | 10.0,1.0,10.0 |
| | | Number of bounding boxes | 30 |
| | Diffusion | Noise schedule | Square Cosine Schedule |
| | | Noise coefficient | 0.0001, 0.02 |
| | | Number of forward diffusion steps | 100 |
| | | Number of reverse denoising steps | 2 |

Predictive Driver Model Score (PDMS) as a comprehensive performance metric, which integrates performance across multiple key driving dimensions through weighted integration, including No at-fault Collision (NC), Drivable Area Compliance (DAC), Time-To-Collision (TTC), Comfort (C), and Ego Progress (EP). The detailed calculation methods for each metric are as follows:

The PDMS used in this study comprehensively evaluates autonomous driving policy performance by integrating sub-metrics across multiple key driving dimensions. Below is a detailed explanation of the definition and calculation method for each sub-metric:

- **No at-fault Collision**: Measures whether the ego vehicle has responsibility collisions with other traffic participants during simulation. The calculation adopts a tiered penalty mechanism: if a responsibility collision occurs (such as active collision in dynamic scenarios), then $\text{score}_{NC} = 0$, resulting in a PDMS of 0 for the entire scenario; if there is a collision with static objects (such as stationary vehicles), then $\text{score}_{NC} = 0.5$; if the ego vehicle is stationary or is rear-ended by a vehicle from behind (non-responsibility collision), no penalty is counted, $\text{score}_{NC} = 1$.

- **Drivable Area Compliance**: Evaluates whether the ego vehicle always remains within the drivable area. If the ego vehicle deviates from the drivable area, then $\text{score}_{DAC} = 0$, resulting in a PDMS of 0; if compliant throughout, then $\text{score}_{DAC} = 1$.

- **Time-To-Collision**: Detects whether the minimum collision time between the ego vehicle and other vehicles is below the safety threshold. Default value $\text{score}_{TTC} = 1$; if during the 4-second simulation, the predicted collision time between the ego vehicle and other vehicles is below the threshold (typically 2 seconds), then $\text{score}_{TTC} = 0$.

- **Comfort**: Measures trajectory smoothness, based on thresholds for acceleration and jerk. Points are deducted if acceleration or jerk exceeds predefined thresholds (such as acceleration $\leq 3.0$ m/s$^2$, jerk $\leq 5.0$ m/s$^3$). The final score is calculated by comparing the deviation of actual parameters from thresholds, normalized to the $[0, 1]$ interval.

- **Ego Progress**: Evaluates the efficiency of the ego vehicle's progress along the predetermined path, comparing actual progress with theoretical maximum safe progress. The calculated score $\text{score}_{EP}$ is the ratio of actual progress to theoretical maximum progress (limited between 0 and 1). Theoretical maximum progress is calculated by the PDM-Closed planner (based on search strategy for collision-free trajectories) as a safe upper limit; actual progress is the longitudinal movement distance of the ego vehicle along the path centerline during simulation. If the theoretical maximum progress is less than 5 meters, low or negative progress values are ignored.

PDMS integrates these sub-metrics through the following formula:

$$\text{PDMS} = \underbrace{\left( \prod_{m \in \{NC, DAC\}} \text{score}_m \right)}_{\text{penalty term}} \times \underbrace{\left( \frac{5 \cdot \text{score}_{EP} + 5 \cdot \text{score}_{TTC} + 2 \cdot \text{score}_C}{5 + 5 + 2} \right)}_{\text{weighted average term}} \quad (22)$$

The key logic of this metric design includes: (1) Safety first: NC and DAC serve as hard penalty terms, ensuring policy must meet basic safety requirements; (2) Multi-dimensional balance: coordinating driving efficiency, safety, and comfort through weight allocation (EP and TTC have the highest weights, followed by C); (3) Scenario adaptability: filtering simple scenarios (such as stationary or straight driving), focusing on challenging conditions (such as turning, interaction), avoiding metric domination by trivial samples. This metric design aims to address the limitations of traditional displacement error (ADE), providing a simulation-driven multi-dimensional assessment that better reflects actual driving performance.

### B.2.2 Baselines

- *AD-MLP* [58]: A lightweight multi-layer perceptron (MLP) network that uses only vehicle state as input, without a perception module, performing trajectory prediction only.
- *VADv2* [6]: A probability-based planning framework that constructs a vocabulary of 4096 candidate trajectories through trajectory space discretization.
- *UniAD* [21]: The first multi-task end-to-end framework that unifies perception, prediction, and planning tasks, using a query mechanism to achieve module collaboration.
- *LTF* [9]: A pure vision-based end-to-end driving framework that achieves implicit BEV representation through cross-modal attention mechanisms using images, replacing LiDAR input.
- *PARA-Drive* [52]: A fully parallel architecture that simultaneously executes perception, prediction, and planning tasks, improving inference speed by 3 times.
- *TransFuser* [9]: A multi-modal end-to-end framework that fuses camera and LiDAR features, combining cross-modal attention and multi-scale feature interaction.
- *LAW* [32]: A latent world model that enhances spatiotemporal feature learning through future state prediction, jointly optimizing scene understanding and trajectory planning.
- *DRAMA* [56]: A Mamba-based planner combining multi-scale convolutional encoders and state Dropout, using an efficient Mamba-Transformer decoder to generate long trajectory sequences.
- *Hydra-MDP* [33]: A multi-teacher knowledge distillation framework that integrates human demonstrations and rule-based experts (such as traffic lights, lane keeping).
- *DiffusionDrive* [34]: A truncated diffusion model for real-time sampling, reducing denoising steps to 2 through anchor-guided noise initialization while maintaining multi-modal coverage.
- *GoalFlow* [54]: A flow matching model with goal constraints, introducing a feasible region scoring mechanism and shadow trajectory refinement strategy.
- *Hydra-MDP++* [31]: An enhanced version of Hydra-MDP, adopting a V2-99 [30] image encoder and expanding metric supervision (such as traffic lights, lane keeping, comfort).

## C  Visualization Comparison of Different Methods on the Navtest Benchmark

We conducted a systematic comparative analysis of the trajectory generation capabilities of DiffE2E, DiffusionDrive [34], and TransFuser [12] across diverse driving scenarios, with detailed results shown in Figure 8 and Figure 9.

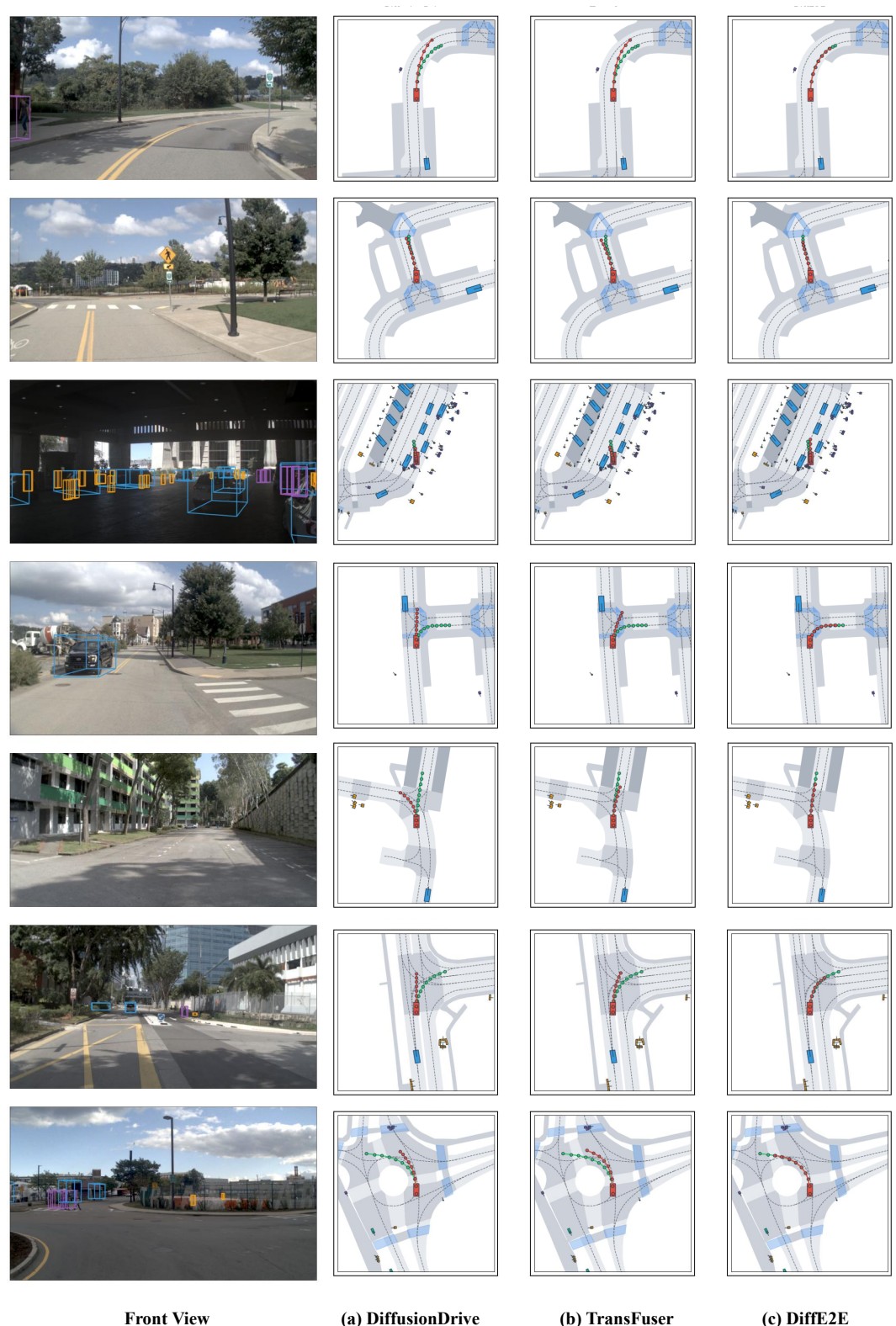

Front View      (a) DiffusionDrive      (b) TransFuser      (c) DiffE2E

Figure 8: Visualization on Navtest benchmark.

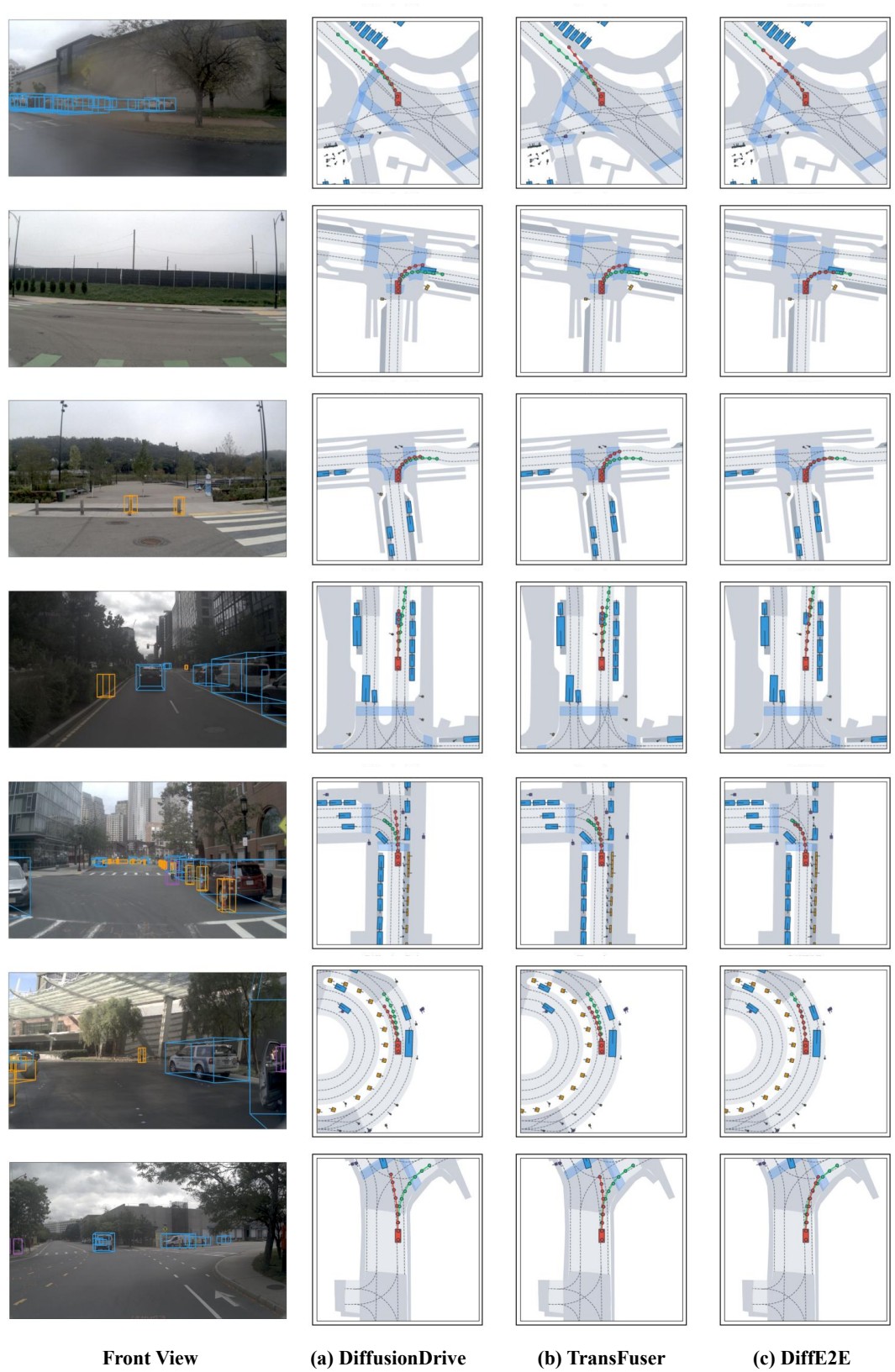

**Front View**   **(a) DiffusionDrive**   **(b) TransFuser**   **(c) DiffE2E**

Figure 9: Visualization on Navtest benchmark.

# D  Additional Studies

In this section, we delve into two key research questions: RQ1: Analysis of the impact mechanism of diffusion model output types, and RQ2: Generalization evaluation in out-of-distribution scenarios. Through a systematic analysis of these questions, we aim to comprehensively reveal the technical advantages and potential limitations of the DiffE2E framework in end-to-end autonomous driving systems.

## D.1  RQ1: Impact Mechanism Analysis of Diffusion Model Output Types

Regarding model output parameterization, we conducted an in-depth study of two strategies for diffusion models: predicting noise versus directly predicting trajectories (original samples), as shown in Table 3. The results reveal a significant disparity—when the model outputs noise, the driving score drops to 20.1, a 76% decrease compared to outputting trajectories, with vehicles barely able to follow lanes in the simulation environment. This substantial difference indicates that directly modeling the trajectory space is more effective than the noise space for tasks requiring high precision, such as autonomous driving.

We attribute this primarily to two factors: First, the trajectory space is inherently structured and semantically rich, with clear temporal correlations, spatial continuity, and a direct mapping to driving task objectives (e.g., safety and efficiency). This allows the model to more effectively capture inherent patterns of driving behavior (such as car following, lane changing, turning, and other typical driving paradigms). In contrast, the noise space lacks clear physical meaning and semantic structure, requiring the model to learn complex mapping relationships from abstract noise to specific trajectories. Second, in the noise prediction paradigm, there is a significant error accumulation effect during the reverse generation process of diffusion models ($x^{(T)} \to x^{(T-1)} \to \cdots \to x^{(0)}$). Small deviations in each denoising step are transmitted and amplified in subsequent steps; as iterations progress, this cumulative error can cause the generated trajectory to deviate severely from the feasible driving space. This is particularly detrimental in tasks requiring high precision like autonomous driving.

## D.2  RQ2: Generalization Evaluation in Out-of-Distribution Scenarios

DiffE2E not only performs excellently on the training set but also demonstrates outstanding generalization ability in out-of-distribution test scenarios across both the CARLA simulator and the NAVSIM benchmark. Particularly in the complex traffic scenarios shown in Figure 3, DiffE2E exhibits highly intelligent contextual adaptation. For instance, when an obstacle vehicle blocks the regular right-turn route in the right lane, the model dynamically adjusts its decision, temporarily planning a safe straight path to avoid the vehicle ahead before completing the turn. This precise response and flexible adjustment to unseen traffic conditions fully demonstrate DiffE2E's robust generalization ability and decision-making intelligence when facing complex and variable driving environments.

This excellent generalization performance stems from the inherent advantages of the diffusion mechanism. During training, DiffE2E effectively builds robust mapping relationships from "non-ideal states" to target trajectories through systematic noise perturbation modeling and gradual denoising learning. This enables the model to handle various perception anomalies, rare behaviors, and unseen scenarios. More importantly, unlike traditional methods, DiffE2E models the complete trajectory probability distribution rather than a single deterministic solution. This distribution-based modeling approach naturally accommodates the inherent multi-modality and uncertainty in autonomous driving tasks, laying a solid foundation for the model's deployment in complex real-world environments.

# E  Limitations & Future Work

In this section, we discuss the main limitations and directions for future research:

- **Sampling Algorithm.** The current DDIM [44] sampling algorithm faces computational efficiency bottlenecks during trajectory generation. Although DiffE2E can generate trajectories with just 2 denoising steps, complex driving scenarios may require additional denoising steps to ensure high-quality and diverse outputs. Additionally, the current generation process in complex traffic scenarios could benefit from more explicit safety and reliability guidance.

*Future work:* We plan to explore more efficient sampling algorithms, such as DPM-Solver [36] and Elucidated Diffusion Models (EDM) [29]. These methods can significantly reduce the number of sampling steps while maintaining generation quality through more precise ODE/SDE numerical approximations. We also aim to incorporate consistency models [46], distillation algorithms [37], or Shortcut Models [16] to further accelerate sampling efficiency.

Furthermore, we will employ classifier-based or energy-based guidance techniques to constrain the sampling process. By integrating prior knowledge of traffic rules and collision avoidance, we expect to enhance the safety and controllability of the generated trajectories. Combining efficient sampling algorithms with domain-specific guidance mechanisms is expected to significantly improve real-time performance while ensuring high trajectory quality.

