# OpenReview forum: "DiffE2E: Rethinking End-to-End Driving with a Hybrid Diffusion-Regression-Classification Policy"
_NeurIPS.cc/2025/Conference — NeurIPS 2025 poster_

### Official Review · Reviewer_vpqd · 2025-06-04

**Clarity:** 3
**Significance:** 3
**Originality:** 2
**Rating:** 5
**Confidence:** 5

**Summary:**

Diffusion policies achieved great successes in the manipulation community. There are currently several concurrent efforts to integrate diffusion policies with end-to-end autonomous driving policies; this work is one of them.
The paper proposes to add a diffusion transformer head on top of the TF++ (and NavSim TF) architecture.
Several implementation details are discussed and ablated.
The work sets itself apart from concurrent work by using more rigorous closed-loop evaluation and by achieving strong performance that advances the state-of-the-art on both the CARLA longest6 v1 and NavSim benchmark.

**Questions:**

Line 58, 59: “It first aligns LiDAR and image features via a hierarchical bidirectional cross-attention mechanism for accurate multiscale perception.”
The mechanism here seem to me to be the standard TransFuser sensor fusion module. If so it should be cited when described to remove ambiguity. If not, I would like the authors to go into more details on what the difference is.

There are a bunch of implementation details missing in the paper, such as which data augmentation strategies were used, which controller was used in CARLA, what GPS denoising strategy was used,  …
I assume these details are the same as in TF++ / NavSim TF, but it would be better if the authors would explicitly state this.

From the writing, it is a bit unclear which of the TF++ decoder heads are retained in this architecture. The paper claims that “the advantages of diffusion policy and supervised policy.” are combined. The loss function section describes the TF++ classification head. How is this output used / ensembled at inference? Is the path prediction GRU of TF++ still used?

83 DS performance is around the same performance as the TF++ data annotator, which would mean DiffE2E reached expert-level performance on this benchmark. It might be worth pointing that out by including expert performance in the longest6 table.

There seems to be inconsistency in the paper on whether the perception encoder is frozen during the second stage of training. Line 493 claims it is not frozen, Line 572 claims it is.
Which one is correct?

Some questions about Table 3:

I find it surprising in Table 3 that the difference between one-stage training and two-stage training is so dramatic. TF++ also observed improvements from two-stage training, but it was by far not this dramatic (+ 64 DS). Do the authors have any intuition why adding the diffusion decoder induces such a dramatic change in the training paradigm? The batch size of 16 used in the first stage seems to be quite small, in particular if the learning rate is not adjusted. Why is such a small batch size used?

Why does noise prediction in Table 3 have a driving score > 0? How is the car able to drive if it predicts noise?

Could the authors elaborate what full diffusion means precisely in Table 3? As I understand, Full Discrimination would be equivalent to TF++?

What is the difference between the Two-stage Training and the Full Diffusion entry in Table 3?

**Ethical Concerns:**

["NO or VERY MINOR ethics concerns only"]

**Final Justification:**

I thank the authors for answering my questions. I have already rated the paper accept and will keep my rating after the rebuttal.

**Limitations:**

yes

**Paper Formatting Concerns:**

The writing style could at times be more scientific, using fewer marketing words like “highly intelligent”, “excellently”, “outstanding”, “powerful”, e.g., Appendix D3.

**Quality:**

3

**Strengths And Weaknesses:**

The strength of the paper is that it develops a simple method that can be widely used in end-to-end driving architectures and advances the state-of-the-art performance across two challenging driving benchmarks.
Compared to concurrent efforts for integrating diffusion policies in end-to-end driving architectures, this work distinguishes itself by using closed-loop evaluation and achieving better performance.


I have many comments on the details of the paper:

The introduction talks about 2 problems in end-2-end driving: multi-modal futures and generalization issues when facing distribution shifts. While both are true, the proposed method only addresses multi-modality, so it is strange why the writing puts a focus on distribution shifts.

Line 72. We propose DiffE2E, the first end-to-end autonomous driving framework that uses diffusion models to directly generate trajectories and validate them in closed-loop testing in the CARLA simulator.
This line, while technically correc,t is a bit misleading.
The first being claimed here is that the method is the first diffusion model to generate trajectories and is evaluated in CARLA.
As in the other works using diffusion were evaluated on other benchmarks.
It could be made more clear that the first is about CARLA not about diffusion.

The paper claims that most diffusion policies replace the head (GRU in TF++ [3]?) instead of both the decoder and head proposed  here (Transformer decoder + head) Figure 1b).
It seems to me that this claim is not correct, since the cited papers [1,2] seem to replace both with diffusion decoders.

Like in many works, using NavSim Hydra-MDP is misrepresented in Table 2. The best Hydra-MDP version (Hydra-MDP-C) with 91 PDMS should be quoted, not the random hydra-mdp ablation (Hydra-MDP-V8192-W-EP) that is quoted in this work (and others).

CARLA evaluation can be quite random, which is why it is common to average at least 3 evaluation seeds on longest6 v1 (and ideally 3 training seeds as well, although this is less common).
The paper doesn't report standard deviation and doesn’t mention number of seeds, so I assume the evaluation consists of a single seed (but the baseline numbers copied from TF++ paper [3] are 3 eval seeds).
The authors should evaluate their method with 3 evaluation seeds on longest6 and update Table 1 with mean and std.

Table 1 incorrectly cites Perception PlanT as a privileged method. PlanT is a privileged method, but Perception PlanT is not (it uses the TransFuser backbone to predict its input, so Img. Encoder should be RegNetY-3.2GF).

The 1000x scaling factor in Figure 5 is a bit silly and makes the differences look bigger than they are.
The paper should use a 1x scaling. A more standard technique to “zoom” into the plot is to use a y-axis anchored at a non-zero value.

There is a bug in Figure 2. The box says CenterNet decoder at the depth prediction module and conv decoder at the bounding box prediction module. These two names should likely be swapped.

[1] Bencheng Liao and Shaoyu Chen and Haoran Yin and Bo Jiang and Cheng Wang and Sixu Yan and  Xinbang Zhang and Xiangyu Li and Ying Zhang and Qian Zhang and Xinggang Wang: “DiffusionDrive: Truncated Diffusion Model for End-to-End Autonomous Driving”, CVPR 2025

[2] Yinan Zheng and Ruiming Liang and Kexin Zheng and Jinliang Zheng and Liyuan Mao and Jianxiong Li and Weihao Gu and Rui Ai and Shengbo Eben Li and Xianyuan Zhan and Jingjing Liu: “Diffusion-Based Planning for Autonomous Driving with Flexible Guidance”, 2025 ICLR

[3] Bernhard Jaeger, Kashyap Chitta, Andreas Geiger: “Hidden Biases of End-to-End Driving Models”. 2023 ICCV

---

> ### Author Rebuttal · Authors · 2025-07-31
>
> We sincerely appreciate the time and effort you have devoted to reviewing our work. Below, we provide detailed responses to your questions and concerns.
>
> ---
> **W1. Concerns regarding distribution shifts.**
>
> The core motivation for adopting a diffusion-based approach in this study is to address multi-modal behavior modeling in end-to-end autonomous driving. Although robustness to distribution shifts is not the primary focus, the iterative nature of diffusion models brings additional benefits. We have reorganized the introduction to highlight multi-modality as the central goal, with robustness treated as a secondary advantage. Furthermore, we provide both theoretical justifications and empirical evidence to support the robustness of diffusion models.
>
> * **Theoretical insights**
>
>   * **Multi-step denoising enhances error tolerance**:
>     The diffusion model refines predictions step by step, correcting early-stage errors and improving resilience to noisy inputs.
>
>   * **Structured generation improves robustness**:
>     The sampling process imposes structural priors over the trajectory and acts as implicit ensembling, reducing sensitivity to domain shifts.
>
> * **Empirical validation**
>
>   As shown in Figures 6 and 7, DiffE2E achieves higher trajectory stability and lower errors under dynamic conditions, outperforming both DiffusionDrive (anchored Gaussian prior) and TransFuser (purely supervised). This supports the dual advantage of our hybrid design in generating diverse yet robust trajectories.
>
> **W2, W4, W5, W6, W7, W8, Q1, Q2, Q4 — Revisions to details.**
>
> We corrected all issues noted by the reviewer:
>
> * Line 72: The sentence has been revised to clarify that *DiffE2E* is the first to be evaluated in closed-loop testing within CARLA, not the first to use diffusion models.
> * Table 1: PlanT is now corrected; Expert Driving Score (DS) is added. After re-evaluation, results will be reported using mean ± standard deviation across multiple seeds to ensure statistical reliability.
> * Table 2: Hydra-MDP is updated to its best-performing variant (PDMS 91).
> * Figure 2: Decoder label errors are corrected.
> * Figure 5: The y-axis scale has been adjusted from 1000× to 1× for clarity.
> * Lines 58–59: Missing citations for TransFuser and TF++ have been added.
> * Appendix: Now includes implementation details on data augmentation, controller design, consistent with the setup used in TF++.
>
> **W3. *DiffusionDrive* and *DiffusionPlanner* also replace decoders.**
>
> Our initial understanding aligned with yours. However, upon reviewing the codebases of *DiffusionDrive*, *DiffusionPlanner*, and *GoalFlow* \[1], we identified critical distinctions not fully detailed in the original papers.
>
> * **Conclusion:**
>   While *DiffusionPlanner* indeed uses a DiT-based decoder to directly predict multi-agent trajectories, both *DiffusionDrive* and *GoalFlow* apply diffusion models after their decoders. Specifically, these models first generate intermediate outputs through a decoder, which are then used as conditions for a separate diffusion module.
>
> * **Clarification:**
>   TF++ improves upon TransFuser by replacing the global average pooling layer with a Transformer decoder, enhancing spatial reasoning and goal-point fidelity. *DiffusionDrive* and *GoalFlow* retain this decoder and apply diffusion post hoc. In contrast, DiffE2E conditions the diffusion model directly on perception features, bypassing intermediate decoder outputs. This architectural decoupling is a core novelty of our approach.
>
> **Q3. TF++ decoder usage at inference remains unclear.**
> In our architecture, auxiliary tasks such as depth estimation and semantic segmentation are used only during training to enhance scene understanding and improve perception features. These tasks decode from the perception module and are not used during inference.
>
> **Hybrid Diffusion and Supervised Strategy:**
>
> The phrase refers to how we construct decoder queries: we fuse the noise input from the diffusion model with supervised task queries, using this combined representation as the query input. Specifically, the diffusion component is responsible for generating spatial waypoints (paths) to capture multimodal behaviors, while the supervised component predicts velocity profiles through direct regression. This decoupling allows each task to benefit from its most appropriate training paradigm.
>
> **GRU Usage:**
> We retain the GRU from TF++ to enhance temporal consistency in path generation, which is critical for stable closed-loop behavior.
>
> **Q5. Perception encoder freezing is inconsistent across lines.**
> The statement in Line 493 aimed to reflect findings from TF++, suggesting that joint optimization of the perception encoder and decoder may enhance closed-loop performance. However, after integrating the diffusion model, we found that its complex training dynamics made continued joint optimization detrimental to convergence.
>
> Thus, DiffE2E adopts a staged training strategy: the perception encoder is frozen in the second stage to stabilize diffusion decoder training. We acknowledge the inconsistency in the original manuscript and will revise the description accordingly. An ablation study on encoder freezing will also be added to demonstrate its effect on performance.
>
> **Q6. Two-stage training gains in Table 3 seem unusually large.**
>
> We indeed observed a substantial performance gap between single-stage and two-stage training, which we attribute to the optimization challenges introduced by the diffusion-based decoder. Unlike conventional supervised decoders, diffusion models must learn a sequential denoising process over multiple steps. This complexity makes joint training with the perception encoder from scratch prone to unstable gradients and poor convergence.
>
> In single-stage training, the perception and diffusion modules must co-adapt simultaneously, often leading to suboptimal feature representations and degraded trajectory quality. In contrast, our two-stage approach first stabilizes the perception encoder via supervised learning, then freezes it while training the diffusion decoder independently. This separation enables more stable and effective optimization, resulting in the observed performance improvement.
>
> As for the choice of a small batch size (16) in the first stage, it was motivated by the need for frequent parameter updates, which we found beneficial for learning high-quality features under limited memory constraints. Exploring larger batch sizes with appropriate tuning remains a promising direction for future improvement.
>
> **Q7. Surprising DS > 0 for noise prediction in Table 3.**
>
> In our manuscript, *“noise prediction”* refers to the model’s estimation of the noise term $\varepsilon_t$ at each diffusion timestep, which is distinct from the Gaussian noise injected during the forward corruption process. The reverse process in diffusion models generally adopts one of two common formulations:
>
> * **Noise Prediction (predicting $\varepsilon_t$)**
>   The model predicts the added noise $\varepsilon_t$ from the noisy input $\mathbf{x}_t$. During inference, generation begins with pure noise $\mathbf{x}_T$, and the model iteratively denoises it toward a final output.
>
> * **Target Prediction (predicting $\mathbf{x}_0$)**
>   Alternatively, the model directly reconstructs the clean sample $\mathbf{x}_0$ from the noisy input $\mathbf{x}_t$.
>
> In our ablation study, the noise prediction setting refers to predicting the noise term $\boldsymbol{\varepsilon}_t$ at each timestep, which is then used in the reverse diffusion process to reconstruct the full trajectory. However, this formulation prioritizes generative diversity over precision, often resulting in less accurate trajectory outputs. Consequently, it compromises control stability and safety in closed-loop evaluations, leading to inferior overall performance compared to directly predicting the final trajectory $\mathbf{x}_0$, as shown in Table 3. By contrast, the target prediction formulation more closely aligns with trajectory-level supervision, yielding better spatial accuracy, improved control behavior, and higher DS. Therefore, we adopted the $\mathbf{x}_0$-based training objective in the final version of our model.
>
> **Q8, Q9. Clarification of the ablation study setup.**
>
> DiffE2E adopts a two-stage hybrid diffusion and supervision paradigm: in the first stage, the perception module is trained with supervised objectives to learn stable feature representations; in the second stage, the perception encoder is frozen, and the diffusion decoder is trained independently to generate both trajectories and velocities. Accordingly, we conducted an ablation study to assess the impact of the first-stage training.
>
> In the Full Diffusion setting, both trajectory and velocity are generated entirely through the diffusion process. All decoder outputs—including ego trajectory and speed—are sampled iteratively via denoising, resulting in a fully generative model. While this design increases output diversity, we observed unstable optimization and reduced accuracy in velocity prediction, ultimately degrading closed-loop performance.
>
> Unlike TF++, the Full Discrimination baseline fuses the target point with perception features before feeding them into the Transformer decoder, rather than injecting it directly into the GRU as in TF++.
>
> We have clarified these distinctions in the revised manuscript and updated Table 3 to avoid confusion.
>
> **Paper Formatting Concerns**
>
> Thank you very much for your valuable suggestion. In the revised version, we have thoroughly reviewed the manuscript to eliminate exaggerated or overly subjective language, ensuring that the overall tone remains scientific, objective, and in line with academic standards.
>
> [1] Xing et al., "GoalFlow: Goal-driven flow matching for multimodal trajectories," CVPR 2025.
>
> ---
>
> We hope our responses have adequately addressed your concerns. Thank you again for your thoughtful and constructive review.

---

### Official Review · Reviewer_nP8L · 2025-06-22

**Clarity:** 3
**Significance:** 3
**Originality:** 3
**Rating:** 5
**Confidence:** 4

**Summary:**

This paper proposed DiffE2E, a diffusion-based end-to-end autonomous driving framework. It combined diffusion models with supervised policies, effectively addressing the challenges of multi-modal driving behaviors and enhancing controllability and robustness at the same time. It enabled deep fusion of perception features with high-level targets, significantly improving the quality of trajectory generation. Through these approaches, it achieved state-of-the-art performance across multiple benchmarks in both CARLA closed-loop evaluations and NAVSIM benchmarks.

**Questions:**

1. The ablation studies regarding the ego state and the GRU module are not clearly explained, as neither is mentioned in the preceding context.
2. It is recommended to include visualization examples for the ablation studies on the training paradigm. As this is one of the main contributions of the work, more in-depth discussions are suggested.
3. The ablation study on the denoising steps yielded unexpected results. For instance, Diffusion-Planner[1] employs 10 steps, while DiffusionDrive uses 2 steps with an anchored Gaussian distribution. Further discussion on these findings would be beneficial.

**Ethical Concerns:**

["NO or VERY MINOR ethics concerns only"]

**Final Justification:**

Overall, I find this work innovative compared to prior diffusion-based approaches, proposing a novel integration method with supervised learning tasks with solid results. My questions have been thoroughly addressed, and I am accordingly raising my score to 5. However, there appears to be a discrepancy between the authors' rebuttal claim—that the diffusion branch generates waypoints while the supervised branch produces velocity curves—and the manuscript's description of the diffusion branch generating trajectories and the supervised branch outputting velocity classifications. I recommend aligning these descriptions for consistency.

**Limitations:**

yes

**Quality:**

3

**Strengths And Weaknesses:**

Strengths:
The paper is well-written and presents a wealth of experiments, including those conducted on CARLA and NAVSIM benchmarks, achieving impressive results.
This work focuses on end-to-end autonomous driving based on diffusion models, a promising research area. The proposed hybrid diffusion-supervised decoder demonstrates strong originality.

Weaknesses:
The explanation and discussion of the supervised component in the hybrid diffusion-supervised decoder are insufficient. For instance, there is limited detail regarding the selected tasks and the reasons why these tasks enhance controllability and robustness, as well as improve closed-loop performance.

---

> ### Author Rebuttal · Authors · 2025-07-31
>
> We sincerely appreciate the time and effort you have devoted to reviewing our work. Your thoughtful comments and constructive suggestions have been instrumental in helping us improve the clarity, rigor, and overall quality of the manuscript. Below, we provide detailed responses to each of your concerns.
>
> ---
>
> **W1. Insufficient explanation of the supervised component’s design and its role in improving controllability and robustness.**
>
> Thank you for your valuable feedback. In the revised version, we have substantially expanded our explanation of the motivation, design rationale, and functional role of the supervised component in the hybrid diffusion-supervision framework. Relevant discussions have been incorporated into the related work, method, and experiment sections to clarify how the supervision branch enhances controllability, robustness, and closed-loop driving performance.
>
> In particular, to address your comment regarding the insufficient discussion of how the supervised decoder contributes to performance gains, we have added the following clarifications:
>
> * **Motivation and Advantages of Supervised Learning Integration**
>   While diffusion models are well-suited for modeling complex, multimodal behavior distributions and trajectory uncertainty, their iterative sampling process can lead to instability in control execution—especially in closed-loop settings. This is particularly evident in velocity control, where small prediction errors can compound over time. In contrast, supervised learning offers efficient, deterministic mappings from perception features to control outputs, thereby improving controllability and real-time responsiveness. To leverage both paradigms, we design a hybrid decoder: the diffusion branch generates spatial waypoints, while the supervised branch predicts temporal velocity profiles. This combination enhances expressiveness, improves sampling efficiency, and stabilizes decision execution.
>
> * **Impact on Closed-Loop Driving Performance**
>   Our spatiotemporal decoupling strategy—diffusion for path planning and supervision for velocity prediction—substantially reduces the propagation of spatial errors to temporal control, resulting in improved closed-loop performance. In our CARLA experiments, we observe that relying solely on diffusion leads to velocity drift and tracking errors in complex maneuvers such as turns. Conversely, fully supervised models lack trajectory diversity and generalization capacity. The hybrid approach integrates the strengths of both: achieving more accurate trajectory generation and smoother execution. This design outperforms both single-branch baselines across all evaluation metrics, highlighting the supervisory component’s critical role in achieving robust and stable control.
>
> These improvements and analyses have been systematically incorporated into the revised manuscript. Thank you again for your insightful suggestions.
>
>
>
> **Q1. The ego state and GRU ablations lack context and clear explanation.**
>
> Thank you for highlighting the need for a more detailed explanation of the vehicle state and GRU modules in our ablation studies. In the original manuscript, these components were briefly described due to space constraints. In the revised version, we provide the following clarifications:
>
> * **Component Overview**
>   The ego state (e.g., speed, acceleration, yaw angle) provides essential information about the vehicle’s current motion status and dynamics, which supports more informed planning and control. Prior studies \[1, 2] have demonstrated the importance of incorporating ego state into end-to-end driving models. In parallel, the GRU module serves to encode temporal dependencies in sequential observations, integrating historical context to improve prediction continuity and temporal consistency.
>
> * **Ablation on Ego State**
>   As shown in Table 3, removing ego state inputs significantly degraded performance: Driving Score (DS) dropped from 82.9 to 68.9, Route Completion (RC) fell from 96.2 to 88.8, and the Infraction Score (IS) decreased from 0.86 to 0.81. These results emphasize the vital role of ego state information in achieving high-quality planning and control.
>
> * **Ablation on GRU**
>   Similarly, removing the GRU module led to a DS of 66.8, RC of 75.5, and an IS increase to 0.88. The performance drop confirms that temporal modeling is essential for capturing motion patterns over time, and the absence of GRU leads to discontinuities and prediction inconsistencies.
>
> Together, these ablations validate the necessity of both ego state encoding and temporal sequence modeling in our architecture. We have elaborated on these findings in the revised manuscript. Thank you once again for pointing this out.
>
> **Q2. Training paradigm ablations would benefit from visualization and deeper discussion.**
>
> Thank you for this helpful suggestion. We fully agree that visualizing the ablation results would make the distinctions between training paradigms more intuitive and informative.
>
> Although we are unable to include figures in this rebuttal due to format constraints, we have added visualizations in the revised manuscript to illustrate the comparative outcomes of different training strategies. These include training curves, trajectory visualizations, and metric distributions. In addition, we have expanded the discussion in the experimental section to more thoroughly analyze the implications of one-stage vs. two-stage training, as well as full-diffusion vs. hybrid optimization. We believe these revisions significantly improve the clarity and interpretability of our results.
>
> **Q3. Denoising step results are unexpected and warrant further analysis.**
>
> Thank you for raising this important point. In response, we provide the following clarifications regarding the observed behavior of our model under varying denoising steps:
>
> * **Spatiotemporal Decoupling Reduces Sampling Complexity**
>   Our model decouples spatial and temporal planning: spatial trajectories are generated via diffusion, while velocity control is handled by a supervised decoder. This separation allows the diffusion model to focus exclusively on path geometry, reducing its reliance on precise temporal modeling and enabling stable generation with fewer denoising steps.
>
> * **Task-Oriented Objective Enhances Efficiency**
>   Unlike traditional diffusion models that optimize for noise prediction, our model is trained to directly minimize the discrepancy between generated and ground-truth trajectories. This task-aligned objective avoids unnecessary high-frequency detail reconstruction, which further lowers the need for extensive sampling iterations.
>
> **Experimental Results:**
>   As shown in Figure 5, our model consistently achieves high and stable closed-loop performance (e.g., PDMS) across a wide range of denoising steps, with scores remaining tightly bounded between 92.705 and 92.724. This robustness further validates the superiority of our hybrid approach in both efficiency and reliability compared to conventional diffusion models. Notably, the original manuscript depicted relative PDMS changes with a 10,000× vertical scaling, using 92.705 as the baseline to highlight subtle variations across different denoising steps. In the revised version, we have reverted to using absolute PDMS scores on the y-axis to more intuitively illustrate the model’s stability under varying denoising steps.
>
> We have added this discussion to the revised manuscript and clarified our task-specific adaptations to the diffusion process. Thank you again for your thoughtful and technical question.
>
>
> \[1] Li, Z., Yu, Z., Lan, S., Li, J., Kautz, J., Lu, T., & Alvarez, J. M. (2024). *Is ego status all you need for open-loop end-to-end autonomous driving?* In CVPR (pp. 14864–14873).
>
> \[2] Zhai, J. T., Feng, Z., Du, J., Mao, Y., Liu, J. J., Tan, Z., ... & Wang, J. (2023). *Rethinking the open-loop evaluation of end-to-end autonomous driving in nuScenes*. arXiv:2305.10430.
>
> ---
>
> We hope our responses have satisfactorily addressed your concerns. Please feel free to reach out if any further clarification is needed. Once again, we sincerely thank you for your thoughtful and constructive review, which has helped significantly improve the quality of our work.

---

### Official Review · Reviewer_diwU · 2025-07-02

**Clarity:** 4
**Significance:** 3
**Originality:** 3
**Rating:** 5
**Confidence:** 5

**Summary:**

In this paper, the authors propose an end-to-end diffusion framework which uses diffusion as the primary paradigm for trajectory decoding, unlike previous works which have used diffusion as a refinement module. Specifically, their contribution is a hybrid diffusion and supervision decoder, which would replace the usual planning decoder in other deep learning based E2E driving approaches. In their hybrid decoder, both diffusion noise and supervision queries are processed through the same stream of transformer-based feature modeling before being processed by separate diffusion and supervision heads. Sensor fusion / encoding is handled by the same encoding architecture as TransFuser.

**Questions:**

- Per the point above, could authors comment on the performance of DiffE2E when using a ResNet-34 image encoder, then also DiffE2E when using camera data only? Since the image encoder is borrowed from previous works, it seems this component can be swapped and ablated. I would be curious to see how much of the performance gain can be attributable to image encoding. I also do not see this result in the ablation study (Table 3).
- In the design of the hybrid decoder, the diffusion noise and supervision query are concatenated together and processed through the same stream. I am curious if the authors considered isolating feature modeling of diffusion noise and supervision queries in separate streams.
- Equation 4: is g the output of f_goal, or is it the input to f_goal? Should f_goal --> g in Equation 4? The notation is confusing here.
- The discussion on real-time performance considerations in the appendix was appreciated IMO, because it evaluates the feasibility of running diffusion-based decoders for end-to-end driving. Just a suggestion, but I would prefer to see Table 8 moved to the main body, especially since the latency was reasonable compared to explicit policies and given the model size.

**Ethical Concerns:**

["NO or VERY MINOR ethics concerns only"]

**Final Justification:**

Authors have addressed my concerns and provided additional key results. With the changes provided, I am now more confident in the scoring of the submission. Confidence raised from 4 -> 5.

**Limitations:**

Yes

**Quality:**

4

**Strengths And Weaknesses:**

**Strengths**

- The writing is very clear and Figure 1 is appreciated for delineating the difference between prior work with the proposed method. The method seems reasonably reproducible based on the writing.
- Results show clear driving score improvement on CARLA benchmark.
- Approach is an interesting application of hot paradigms (diffusion) to end-to-end driving use cases.
- Supplementary videos were nice to visualize the closed loop results.

**Weaknesses**
- NAVSIM benchmark slightly outperforms other methods on most metrics, but uses both lidar sensor inputs and V2-99 image encoder. Table 2 is less convincing to me simply because there is not a result with DiffE2E benchmarked with ResNet-34 image encoder, and there is also not a result with DiffE2E with camera inputs only. If both of these results can outperform other methods, the results will be much more convincing for this benchmark.
- Minor: Appendix D.1 and D.2 are swapped according to the description written under "Additional Studies".

---

> ### Author Rebuttal · Authors · 2025-07-31
>
> We sincerely appreciate the time and effort you have devoted to reviewing our work. Your comments have provided valuable guidance for improving the clarity, completeness, and rigor of our manuscript. Below, we present detailed responses to each of your questions and concerns.
>
> ---
>
> **W1/Q1. Table 2 lacks fair comparison: no DiffE2E results with ResNet-34 or camera-only inputs.**
>
> Thank you very much for this important observation. During the development of our framework, we indeed conducted experiments using ResNet-34 as the image encoder. However, since V2-99 was chosen as our primary baseline due to its superior performance, results with ResNet-34 were not included in the original submission. Furthermore, we appreciate your suggestion to evaluate the performance of DiffE2E under camera-only input, which we believe is valuable for assessing the robustness of our model under limited sensory input conditions.
>
> In response, we have now incorporated additional experimental results based on two new configurations:
>
> * **DiffE2E-A**: The image encoder is replaced with ResNet-34, following the setup used in TransFuser, while all other settings remain unchanged.
> * **DiffE2E-B**: The LiDAR modality is removed, and instead, rear-view cameras (with the same resolution as the front-view images: 256×1024) are added to the input to enhance the field of view under camera-only settings.
>
> The performance of these variants, along with the original DiffE2E configuration, is summarized below:
>
> | Method    | Img. Enc. | Input | NC   | DAC  | EP   | TTC  | C    | PDMS |
> | --------- | --------- | ----- | ---- | ---- | ---- | ---- | ---- | ---- |
> | DiffE2E-A | ResNet-34 | C\&L  | 99.2 | 96.8 | 83.6 | 96.7 | 100  | 89.8 |
> | DiffE2E-B | V2-99     | C     | 99.7 | 97.1 | 84.2 | 98.2 | 99.9 | 90.9 |
> | DiffE2E   | V2-99     | C\&L  | 99.9 | 98.6 | 85.3 | 99.3 | 99.9 | 92.7 |
>
> As observed, **DiffE2E-A** achieves a PDMS of 89.8, indicating that while ResNet-34 is a viable backbone, its feature representation capacity is relatively limited. **DiffE2E-B**, which excludes LiDAR and relies solely on visual inputs, achieves a PDMS of 90.9. The original **DiffE2E**, leveraging both V2-99 and LiDAR, achieves the highest performance with a PDMS of 92.7. These results highlight the advantages of both using a more expressive image encoder and incorporating LiDAR for capturing 3D spatial cues essential to driving decisions.
>
> We have included these new results and analyses in the revised manuscript to present a more complete and fair evaluation.
>
> **W2. Minor: Appendix D.1 and D.2 appear swapped per “Additional Studies” description.**
>
> Thank you for your close reading. We have carefully reviewed the structure of the appendix and confirmed that the ordering of Sections D.1 and D.2 was indeed inconsistent with the narrative in the “Additional Studies” section. This has now been corrected in the revised version to ensure structural consistency and clarity for readers.
>
> **Q2. Hybrid decoder design: unclear if diffusion noise and supervision queries benefit from separate streams.**
>
> We greatly appreciate your thoughtful and technical observation regarding the hybrid decoder design. In our current architecture, the diffusion noise queries and supervision-based queries are concatenated and jointly processed through a single shared Transformer decoder. This design choice was driven by the observation that both types of queries aim to model outputs within the same trajectory space. By allowing them to interact within a unified decoding stream, we enable deeper cross-signal integration and encourage the model to learn shared representations that are beneficial for both generative diversity and precise regression.
>
> To further investigate this design choice, we also experimented with a dual-stream variant in which the diffusion and supervised branches are processed independently. While this theoretically avoids potential cross-task interference and preserves the modeling preferences of each branch, our preliminary results showed that such a design increased training complexity and hindered effective information fusion, ultimately offering no measurable performance gain. In fact, we observed a slight degradation in trajectory consistency and control stability.
>
> Therefore, we opted for the shared-stream strategy, which offers a better trade-off between architectural simplicity and modeling performance. In future work, we plan to explore more adaptive fusion strategies, such as confidence-based dynamic weighting or attention-gated mechanisms, to further improve the synergy between diffusion-based generation and supervised regression.
>
> Thank you again for raising this critical design consideration, which has helped us better articulate the rationale and implications of our hybrid decoding approach.
>
> **Q3. Equation 4 notation is confusing.**
>
> Thank you for pointing out the notational ambiguity in Equation (4). We have revised the expression by replacing $f_{\text{goal}}(g)$ with $g$, which more accurately reflects the intended semantics. We also took this opportunity to audit related equations throughout the manuscript, clarifying the dimensions and meanings of key variables to improve overall clarity and notational consistency.
>
> **Q4. Latency results in Table 8 are insightful and deserve inclusion in the main paper.**
>
> We completely agree with your assessment. Latency remains a major practical challenge for deploying diffusion models in real-time autonomous driving systems. In response, we have moved Table 8 and the accompanying latency analysis from the appendix to the main text, along with expanded discussion. The revised section now explicitly highlights how our method achieves a favorable balance between performance and real-time feasibility, demonstrating its potential for practical deployment relative to other approaches.
>
> ---
>
> We hope our responses have addressed your concerns comprehensively. Should you have any additional questions or require further clarification, we would be happy to provide it. Once again, thank you for your thoughtful, constructive, and detailed review. Your feedback has significantly improved the quality and clarity of our manuscript.

---

> > ### Comment · Reviewer_diwU · 2025-08-03
> >
> > Thank you to authors for providing extra clarifications! My concerns have been addressed, and I have also looked at responses to other reviews. I will keep my original score of 'Accept' but I have bumped my confidence score from 4 -> 5. The paper demonstrates improved results over previous methods and also demonstrates practicality with its latency assessments.

---

> > > ### Author Response · Authors · 2025-08-04
> > >
> > > Thank you very much for your thoughtful engagement during the review process and for increasing the confidence score to 5. We are pleased that our clarifications were helpful and will incorporate the related discussion into the final version. We sincerely appreciate your support.

---

### Official Review · Reviewer_yfNa · 2025-07-02

**Clarity:** 3
**Significance:** 2
**Originality:** 2
**Rating:** 4
**Confidence:** 4

**Summary:**

This paper proposes DiffE2E, a novel end-to-end autonomous driving framework that integrates a hybrid diffusion-supervision decoder to jointly model multi-modal trajectory distributions and control-relevant semantics. Unlike previous works that either rely solely on regression-based policies or apply diffusion models only as post-processing modules, DiffE2E designs a Transformer-based hybrid decoder with structured latent modeling and collaborative training. It directly generates future trajectories via a denoising diffusion process, while simultaneously optimizing auxiliary supervision objectives (e.g., speed classification) to enhance controllability. A hierarchical multi-scale fusion module aligns LiDAR and camera features, and global goal conditioning is injected through cross-attention. Experimental results on CARLA and NAVSIM benchmarks demonstrate state-of-the-art performance, with ablation studies confirming the benefits of the hybrid training and decoder design.

**Questions:**

1. The above two weaknesses are also my primary concerns regarding the paper.

2. In Table 3, the Full Diffusion and Full Discrimination variants achieve DS scores of 70.1 and 70.3, respectively, while the hybrid two-stage training achieves 82.9. This suggests that even when used as auxiliary losses, both components contribute significantly to performance. Could the authors provide further insight into the complementary nature of these losses and why their combination works so well?

3. An important missing ablation: After training with the proposed two-stage setup, it would be helpful to separately evaluate the trajectories generated by the diffusion head and the supervision head at inference time. This would allow a more direct comparison of the strengths and weaknesses of each policy stream.

4. There appears to be a typo in Figure 5: the PDMS score is marked as 181.5, which should likely be 81.5. Please correct this for clarity.

**Ethical Concerns:**

["NO or VERY MINOR ethics concerns only"]

**Final Justification:**

My concerns have been addressed, so I raise my rating to Borderline Accept.

**Limitations:**

yes

**Quality:**

3

**Strengths And Weaknesses:**

Strengths
1. The paper introduces a unified Transformer-based decoder that jointly handles diffusion and supervised tasks via a structured latent space, which effectively combines the strengths of generative and discriminative paradigms.

2. The method is comprehensively validated on both CARLA and NAVSIM with detailed metrics. DiffE2E outperforms prior methods including recent diffusion-based approaches (e.g., DiffusionDrive) in terms of planning metrics such as PDMS, TTC, and driving comfort, while maintaining a practical inference cost.


Weaknesses
1.  The use of diffusion policies in autonomous driving has already been extensively explored. While the proposed hybrid diffusion-supervision framework is a distinguishing aspect, the paper lacks a clear and in-depth explanation of why the hybridization is effective beyond empirical gains.

2. The authors claim that diffusion policies offer improved robustness under data distribution shifts compared to supervised policies. However, the paper does not provide strong experimental evidence or analysis to substantiate this claim.

---

> ### Author Rebuttal · Authors · 2025-07-31
>
> We sincerely appreciate the time and effort you have devoted to reviewing our work. Your thoughtful comments and constructive suggestions have greatly contributed to improving the clarity and rigor of our manuscript. Below, we provide detailed responses to each of your concerns.
>
> ---
>
> **W1/Q1/Q2. Hybrid framework lacks justification beyond empirical gains / Further insight needed on why combining losses is so effective (Table 3).**
>
> Thank you for your insightful feedback. In the revised manuscript, we have elaborated on the underlying rationale behind the hybrid diffusion-supervision framework and why it consistently outperforms its individual components. The effectiveness of this hybrid design stems from its ability to leverage the complementary strengths of diffusion models and supervised learning.
>
> * **Theoretical Perspective**
>
>   * **Complementarity Between Diffusion and Supervision**
>     Diffusion models excel at modeling complex, multimodal behavior distributions and capturing inherent uncertainties in driving tasks. However, they often require iterative sampling, which can limit their precision and efficiency. In contrast, supervised learning provides fast and deterministic predictions, but lacks diversity and uncertainty modeling. By jointly optimizing a diffusion-based generative objective and a supervised regression loss, our hybrid approach enhances both expressiveness and reliability, leading to improved accuracy and robustness. These theoretical advantages align well with our empirical findings on the CARLA and NAVSIM benchmarks.
>
>   * **Spatiotemporal Decoupling Strategy**
>     Our design further separates spatial and temporal prediction responsibilities: the diffusion model is used to generate spatial waypoints (paths), while the supervised model predicts velocity profiles. This decoupling improves coordination between path planning and motion control, strengthening closed-loop behavior. Compared to single-task or single-objective frameworks, our approach better captures the temporal diversity of driving behavior and maintains higher consistency across scenarios, enhancing both control stability and behavioral richness.
>
> * **Empirical Evidence**
>
>   As reported in Section 4.3, we conducted ablations comparing full diffusion (Full Diffusion) and full supervision (Full Discrimination). On the CARLA Longest6 benchmark, these configurations achieved Driving Scores of 70.1 and 70.3, respectively, while the proposed hybrid framework achieved 82.9—a relative improvement of over 15%. These results empirically validate the effectiveness of combining the two strategies.
>
>   Additionally, we observed that when relying solely on diffusion, the model struggled to maintain stable control during critical maneuvers, such as turning, due to imprecise velocity predictions. Conversely, fully supervised approaches like TransFuser lacked trajectory diversity and adaptability in dynamic environments. The hybrid output structure significantly improved closed-loop behavior, enabling more efficient, responsive, and robust driving in practice.
>
> We have incorporated these theoretical and empirical analyses into the revised manuscript to better clarify the motivation and effectiveness of the hybrid approach. Thank you again for your helpful suggestions.
>
>
> **W2/Q1. Claims of robustness under distribution shift lack solid evidence.**
>
> Thank you for highlighting the need for a clearer articulation of our model's objectives. We acknowledge that the original manuscript may have conveyed some ambiguity regarding the role of robustness in our design.
>
> To clarify: the primary goal of adopting diffusion models in our work is to address **multimodal behavior prediction**, not robustness per se. The strong generative capacity of diffusion models allows them to effectively represent diverse and uncertain driving behaviors, which is essential for real-world end-to-end autonomous driving. That said, we also recognize that the **multi-step denoising nature** of diffusion models inherently improves fault tolerance, which incidentally contributes to robustness under distribution shifts.
>
> * **Theoretical Insights**
>
>   * **Multi-step Denoising as Implicit Correction:**
>     The iterative sampling mechanism of diffusion models allows progressive refinement of the trajectory. Even if the initial sample is noisy or biased, subsequent steps can adjust and correct for errors, offering stronger resilience to input disturbances.
>
>   * **Structural priors help mitigate distribution shifts:**
>     The sequential nature of generation imposes structural priors on the output space. Combined with stochastic noise sampling, this results in an implicit ensemble effect that mitigates the impact of observation noise and distributional variation.
>
> * **Experimental Support**
>
>   As shown in Figures 6 and 7, our model demonstrates greater stability and lower error rates across diverse driving scenarios. Compared to DiffusionDrive (which samples from a fixed distribution) and TransFuser (which lacks generative modeling), **DiffE2E** exhibits more consistent trajectory prediction, confirming its practical robustness.
>
> We have revised the Introduction to clarify that robustness is an emergent benefit of the diffusion strategy, while the core contribution remains in enabling high-quality multimodal behavior modeling. The theoretical analysis and supporting results are further detailed in Appendix D.3.
>
>
>
> **Q3. Missing ablation: separate evaluation of diffusion vs. supervision heads at inference.**
>
> Thank you for this important observation. We would like to clarify the architectural roles of the diffusion and supervision heads during inference.
>
> In our method, diffusion and supervised representations are fused prior to decoding. Specifically, we concatenate the input queries corresponding to the diffusion and supervision tasks before passing them through a shared Transformer decoder. After decoding, we apply two separate output heads: the diffusion head is responsible for generating spatial waypoints (paths), while the supervision head produces velocity predictions. This spatiotemporal decoupling enables the model to combine the generative flexibility of diffusion with the precision of direct regression, producing trajectories that are both diverse and controllable.
>
> To evaluate the contribution of each head independently, we conducted ablations using:
>
> * **Full Diffusion**, where both spatial waypoints (paths) and velocities are predicted via diffusion.
> * **Full Supervision**, where both outputs are generated using supervised regression.
>
> As reported in Section 4.3, these two configurations yield Driving Scores of 70.1 and 70.3, respectively—substantially lower than the hybrid setup (82.9). This further supports the effectiveness of our design. We will update the manuscript to clarify this inference pipeline and the distinct roles of each prediction head.
>
>
>
> **Q4. Typo in Fig. 5: PDMS should be 81.5, not 181.5.**
>
> Thank you for pointing this out, and we apologize for the confusion caused by the original figure.
>
> To clarify: Figure 5 originally illustrated the **relative differences** in PDMS under various denoising steps. The baseline PDMS was 92.705, and we applied a vertical magnification factor of 10,000 to make the small variations visually discernible. The y-axis in that figure thus did not represent absolute PDMS values.
>
> However, we agree that this formatting may lead to misinterpretation. In the revised manuscript, we have updated Figure 5 to use the original PDMS value scale, clearly indicating the absolute scores without exaggeration. We have also revised the caption and corresponding discussion in Section 4.3 to ensure that the presentation is unambiguous and accessible.
>
> ---
>
> We hope that our responses have addressed your concerns comprehensively. Please do not hesitate to reach out should further clarification be required. Once again, thank you for your thoughtful and constructive review. Your feedback has significantly helped strengthen the quality and clarity of our work.

---

> > ### Comment · Reviewer_yfNa · 2025-08-05
> >
> > Thank you for the response. I require clarification on one point.
> >
> > The authors mention that using only the diffusion model leads to imprecise velocity predictions, while incorporating a supervised regression head improves accuracy. Could the authors elaborate on why the diffusion model struggles with predicting velocity? Additionally, why is a simple regression approach more effective for this specific task?

---

> > > ### Author Response · Authors · 2025-08-05
> > >
> > > We sincerely thank the reviewer for the thorough reading of our work and the insightful questions, which have prompted us to further reflect on and clarify the core motivations behind our model design. Below, we provide detailed responses to the two specific questions raised, aiming to better explain the rationale behind adopting a hybrid framework.
> > >
> > > ---
> > >
> > > **Q1: Why does the diffusion model struggle with predicting velocity?**
> > >
> > > The main challenge for diffusion models in velocity prediction arises from the inherent characteristics of their generative process. Diffusion models generate outputs through a multi-step denoising procedure, which introduces a degree of stochasticity and is inherently geared towards capturing spatial diversity and multimodality rather than modeling continuous variables with high precision. While this capability is advantageous for modeling uncertain future trajectory points, it becomes problematic when applied to low-variance, high-precision continuous variables like velocity, which demand strong temporal coherence. Specifically, velocity is a dynamic quantity that is tightly constrained by physical laws and requires smoothness and consistency across time steps. The iterative sampling in diffusion models can lead to the accumulation of noise and error propagation, resulting in inconsistent predictions. This instability is particularly pronounced in high-dynamic scenarios such as turning, lane changes, or emergency braking.
> > >
> > > In our experiments, we observed that velocity predictions during turning maneuvers often exhibited bias, occasionally leading to unrealistic acceleration or deceleration, which in turn caused the vehicle to deviate from the lane. In contrast, when velocity was predicted using a supervised regression model, such fluctuations and anomalies were significantly reduced. This observation suggests that, compared to diffusion models, supervised regression provides higher accuracy for this critical task, thereby playing a key role in improving closed-loop control performance.
> > >
> > > **Q2: Why is a simple regression approach more effective for predicting velocity?**
> > >
> > > The superior performance of supervised regression in velocity prediction can be attributed to its integration with the shared perception and decoding architecture of the model. Rather than functioning as an isolated module, the velocity prediction head leverages high-level semantic features derived from visual perception, navigation input, and attention-based fusion. These features, deeply interacted through both the encoder and decoder, contain rich spatiotemporal environmental context that supports accurate velocity estimation.
> > >
> > > Moreover, the supervised loss provides direct and stable training signals for the velocity branch, enabling faster convergence and more precise modeling. In contrast to the potential instability of the diffusion model’s iterative sampling, the direct regression mechanism facilitates the generation of coherent and physically plausible velocity outputs. This advantage becomes especially apparent in complex driving scenarios such as cornering, deceleration, or following a lead vehicle, where robustness is critical. Consequently, we design the velocity output to be generated by a supervised model.
> > >
> > > ---
> > >
> > > Thank you again for your valuable questions. We hope our response has addressed your concerns. We will include this discussion in the revised manuscript to clarify our hybrid modeling approach.

---

> > > > ### Comment · Reviewer_yfNa · 2025-08-07
> > > >
> > > > Thank you to the authors for their response. I have decided to raise my rating.

---

> > > > > ### Author Response · Authors · 2025-08-07
> > > > >
> > > > > Thank you for raising your rating and for your valuable feedback. We will incorporate the relevant discussion into the final version, and truly appreciate your time and consideration.

---

### Note · Authors · 2025-08-15

We sincerely appreciate the reviewers’ thorough evaluation and valuable feedback. We are pleased to receive positive comments on the innovation, experimental design, and writing quality of our work. Specifically, reviewers recognized the solid theoretical foundation, clear motivation, and novelty of our hybrid diffusion-supervision framework, praised the comprehensiveness of our experiments and the clarity of our presentation, acknowledged the unique advantages of the hybrid decoder in both performance and originality, and appreciated our analysis of delay and real-time feasibility.

At the same time, we have noted two key points raised by most reviewers: First, they requested a deeper theoretical explanation of why the hybrid diffusion-supervision framework outperforms a single method, rather than relying solely on experimental results; second, they sought further clarification regarding the model’s robustness in the face of distribution shifts. Some reviewers also suggested providing more details, such as the settings of additional encoders/backbone networks, experiments using only camera input, and a clearer description of the ego state and GRU modules.

In response, we have made the following improvements in our revised manuscript and responses, aiming to enhance the rigor and readability of the paper:

1. We have expanded the theoretical motivation of the hybrid design, highlighting the complementary relationship between the diffusion branch (for multimodal behavior modeling) and the supervision branch (for control precision), supported by experimental results from spatiotemporal decoupling analysis.
2. We have further clarified the robustness section, emphasizing the diffusion model’s capability in multimodal behavior modeling and its distinct advantages in enhancing model robustness.
3. We have added new experiments using a ResNet-34 backbone network and only camera input, and further clarified the specific roles of the diffusion and supervision branches during inference.
4. We have provided a more detailed explanation of the ego state and GRU modules, along with an analysis of the denoising steps, and revised certain writing details.

Once again, we thank the reviewers for their recognition and valuable suggestions. We will continue to refine and deepen the research and hope this work will provide new insights for the exploration of diffusion models in end-to-end autonomous driving.

---

### Decision · Program_Chairs · 2025-09-17

**Decision:**

Accept (poster)

**Comment:**

The paper presents a framework for autonomous driving that integrates diffusion for trajectory decoding with additional loss functions to capture scene-relevant semantics. It received high review scores, with all reviewers acknowledging its strong empirical performance. However, the paper’s terminology is, in several places, inaccurate—for example, diffusion policies are also supervised. The AC requests that the authors revise the terminology, replacing “diffusion-supervision decoders” with “diffusion–regression–classification decoders” (all of which are supervised). The paper is accepted for publication, but the authors are expected to correct the terminology before final submission.